

# Six new species of free-living nematodes (Nematoda: Enoplida) from deep-sea cold seeps on Hikurangi Margin, New Zealand

Daniel Leduc

Oceans Centre, National Institute of Water and Atmospheric Research, Wellington, New Zealand

## ABSTRACT

Little is known about the taxonomy of deep-sea nematode species inhabiting cold seep habitats. An opportunity to characterize the nematode species communities of New Zealand cold seeps was provided by a 2019 research voyage to New Zealand's Hikurangi Margin, during which macrofauna cores were obtained at two seeps at approximately 1,250 and 2,000 m water depth. Here, six new species of the order Enoplida are described. *Metacylicolaimus catherinae* sp. nov. represents the first record of the genus for the New Zealand Exclusive Economic Zone and for the deep sea globally. *Halalaimus talaurinus* sp. nov., *Thalassoalaimus duoporus* sp. nov. and *Crenopharynx crassipapilla* sp. nov. are only the second species of their respective genera to be described/recorded from New Zealand waters, and *Oncholaimus adustus* sp. nov. is the eighth species of the genus to be recorded from the region. *Rhabdodemania zealandiaensis* sp. nov. was among the most abundant and widespread species found at the Hikurangi Margin seep sites. A few specimens had been found in a previous ecological study of meiofaunal nematode communities on Chatham Rise, a submarine ridge south of Hikurangi Margin. It is possible that this species has a preference for seep environments due to elevated food availability, however it does not seem to be exclusively found in seeps. We find no evidence for an affinity between nematode seep communities in New Zealand and elsewhere, which is consistent with the high variability in nematode community observed to date among regions. Ongoing work on the ecology and distribution of nematode communities at the Hikurangi Margin seep sites will help determine spatial patterns in abundance and species distributions in more detail, including the identification of any species/taxa with affinities with seeps.

# INTRODUCTION

Nematodes are common in all deep-sea sediment habitats, including cold seeps (*Vanreusel et al., 2010a*), where they can be highly abundant relative to nearby background sites (*Vanreusel et al., 2010b*). Nematode communities at cold seeps are typically characterized by low diversity and are often dominated by a single species; there does not appear, however, to be a typical seep-associated nematode fauna, and high variability in nematode

Corresponding author
Daniel Leduc,
daniel.leduc@niwa.co.nz

community structure has been observed at seeps from different locations (*Vanreusel et al., 2010b*). *Rosli et al. (2016)* provided the first data on the abundance of deep-sea cold seep meiofauna in the New Zealand region. Until recently no data were available on the species composition of nematode communities at New Zealand seeps (*Rosli et al., 2016*, *2018*; *Leduc, 2023*).

Sites of active seepage have been identified on the Hikurangi Margin of New Zealand at depths of 600 to 2,000 m (*Greinert et al., 2010*). The mega-epifaunal communities at these Hikurangi Margin sites are dominated by vesicomyid clams, siboglinid worms and bathymodiolin mussels, which is similar to seeps in other parts of the world, (*Baco et al., 2010*; *Jones et al., 2010*; *Klaucke et al., 2010*). The infaunal communities, however, are characterised by unusually high abundance and biomass of tube-building ampharetid polychaetes derive most of their carbon from aerobic methanotrophy, probably by direct consumption of methane-oxidising bacteria (*Thurber et al., 2010*, *2013*). Whether meiofaunal and/or nematode communities at New Zealand seeps are similar or different to seep communities elsewhere is not yet clear due to lack of species-level data (*Rosli et al., 2016*, *Leduc, 2023*).

In July 2019, a research voyage to New Zealand's Hikurangi Margin took place as part of the research programme "Gas Hydrates: Economic Opportunities and Environmental Implications" (HYDEE) to investigate the potential impact of changes in seafloor methane flux on marine ecosystems. This voyage provided an opportunity to characterize the nematode species communities of New Zealand cold seeps for the first time *(Leduc, 2023)*. Nematodes were obtained from both meiofauna (29 mm diameter, 45 μm mesh size) and macrofauna cores (95 mm diameter, 300 μm mesh size). Here, six new species of the order Enoplida are described from macrofauna cores obtained at two seep sites on Hikurangi Margin. The present study, combined with previously published taxonomic work *(Leduc, 2023)* and ongoing ecological analyses of Hikurangi Margin nematode species distribution data, will help determine the degree of heterogeneity in nematode assemblages among deep-sea cold seeps in New Zealand and will enable broader biogeographic analyses to be conducted.

## MATERIALS AND METHODS

Samples were obtained from the southern end of the Hikurangi subduction margin, off the east coast of New Zealand's North Island, in a region where the Pacific Plate subducts obliquely beneath the Australian Plate. Two cold seep sites were sampled during National Institute of Water and Atmospheric Research (NIWA) cruise TAN1904 (July 2019): Mungaroa at ca. 2,000 m depth, and Uruti South at ca. 1,250 m depth (Table 1). Each site was approximately 1 km in diameter, at the centre of which was an active cold seep of approximately 200–300 m in diameter as determined by the extent of the seafloor backscatter signal for the carbonate rock that forms at the main seepage location. At each site, a video-guided multicorer was deployed at several locations along a transect the centre of the seep site out to approximately 500 m away from the centre of the seep site. The Ocean Instruments MC-800A multicorer was equipped with cores of 9.5 cm internal diameter. The top 5 cm of sediment was fixed in 10% formalin and stained with Rose

**Table 1 Details of cold seep sites and sampling stations (NIWA voyage TAN1904).** Only stations from which nematode specimens are described in the present study are included.

| Seep site | Station | Distance from centre of seep site (m) | Depth (m) | Lat (S) | Long (E) |
|---|---|---|---|---|---|
| Mungaroa | 17 | 87 | 2,070 | 41.9354 | 175.3076 |
| Mungaroa | 19 | 0 | 2,075 | 41.9378 | 175.3112 |
| Mungaroa | 20 | 55 | 2,076 | 41.9378 | 175.3118 |
| Mungaroa | 21 | 149 | 2,077 | 41.9382 | 175.3128 |
| Mungaroa | 23 | 479 | 2,019 | 41.9403 | 175.3157 |
| Uruti South | 61 | 105 | 1,227 | 41.4251 | 176.3510 |
| Uruti South | 64 | 275 | 1,245 | 41.4279 | 176.3485 |
| Uruti South | 66 | 101 | 1,230 | 41.4266 | 176.3497 |
| Uruti South | 67 | 67 | 1,232 | 41.4264 | 176.3500 |
| Uruti South | 68 | 0 | 1,237 | 41.4260 | 176.3506 |
| Uruti South | 70 | 82 | 1,235 | 41.4253 | 176.3509 |

Bengal. Samples were rinsed on a 300 μm sieve to retain macro-infauna, and nematodes were handpicked under a stereomicroscope and transferred to pure glycerol (*Somerfield & Warwick, 1996*).

Species descriptions were made from glycerol mounts using differential interference contrast microscopy and drawings were made with the aid of a camera lucida. Measurements were obtained using an Olympus BX53 compound microscope with cellSens Standard software for digital image analysis. All measurements are in mm (unless stated otherwise), and all curved structures are measured along the arc. The terminology used for describing the arrangement of morphological features such as setae follows *Coomans (1979)*, terminology of stoma structures follows *Decraemer, Coomans & Baldwin (2014)*. Type specimens are held in the NIWA Invertebrate Collection (Wellington). The collection of sediment samples was conducted under Special permit 666 to NIWA granted by New Zealand's Ministry for Primary Industries.

The electronic version of this article in Portable Document Format (PDF) will represent a published work according to the International Commission on Zoological Nomenclature (ICZN), and hence the new names contained in the electronic version are effectively published under that Code from the electronic edition alone. This published work and the nomenclatural acts it contains have been registered in ZooBank, the online registration system for the ICZN. The ZooBank LSIDs (Life Science Identifiers) can be resolved and the associated information viewed through any standard web browser by appending the LSID to the prefix http://zoobank.org/. The LSID for this publication is: urn:lsid:zoobank.org:pub:288A67E3-5436-4F9A-990F-DCFF3E49EE35. The online version of this work is archived and available from the following digital repositories: PeerJ, PubMed Central and CLOCKSS.

## Systematics

Class Enoplea *Inglis, 1983*

Order Enoplida *Baird, 1853*
Family Leptosomatidae *Filipjev, 1916*

**Family diagnosis: (emended from** *Smol, Muthumbi & Sharma (2014)*) Large nematodes
(2–50 mm). Two circles of anterior sensilla: six inner labial sensilla mostly papilliform, six
outer labial and four cephalic sensilla setiform, often the cephalic setae are very short.
Amphids pocket-shaped. Large number of metanemes with caudal filament: dorsolateral
and ventrolateral or only dorsolateral orthometanemes and loxometanemes of type I.
Many species with ocelli. Buccal cavity narrow or wide, sometimes with teeth or tooth-like
thickenings. Pharynx inserts into the body cuticle in the region of the buccal cavity, the
cephalic capsule is variable in form. Three pharyngeal glands open in the buccal cavity.
Pharynx always smooth in outline. Secretory-excretory system, if present, restricted to the
pharyngeal region. Female reproductive system didelphic-amphidelphic with
antidromously reflexed ovaries. Males with two opposed testes. Gonad position relative to
intestine variable in species, with anterior and posterior gonad on opposite sides.
Subventral or ventral precloacal papillae often present. Caudal glands mostly present,
extending into the precaudal region.

**Remarks.** Previous diagnoses of the family stated that tubular precloacal supplements are
never present, however *Metacylicolaimus filicaudatus* (*Ditlevsen, 1926*) *Schuurmans
Stekhoven, 1946* possesses a tubular precloacal supplement.
Subfamily Cylicolaiminae *Platonova, 1970*

**Subfamily diagnosis: (from** *Smol, Muthumbi & Sharma (2014)*) Cephalic capsule
distinct, short and thin-walled, cephalic suture poorly developed and barely noticeable or
distinct with invaginations. Stoma cup- or funnel-shaped, very strongly developed, thick
cuticularized walls and a complex armature consisting of one well-developed dorsal tooth,
two ventrosublateral teeth, small blunt tooth-like processes at the bottom of the stoma and
a serrate plate in the upper part. Amphideal fovea pocket-shaped. Tail pointed, often
filiform.
Genus *Metacylicolaimus Schuurmans Stekhoven, 1946*

**Genus diagnosis: (emended from** *Schuurmans Stekhoven (1946)*) Cephalic capsule
present. Buccal cavity wide, shallow or cylindrical, with a large dorsal tooth, additional
teeth may be present. Lips thick, swollen, outer labial and cephalic setae short. Amphid
pocket-shaped. Spicules hook-shaped and curved. Gubernaculum with dorsal apophysis.
Single tubular or disc-like precloacal supplement may be present. Tail pointed, conical
anteriorly, may be filiform posteriorly.
Type species: *M. filicaudatus* (*Ditlevsen, 1926*) *Schuurmans Stekhoven, 1946*

**Remarks.** Prior to this study, the genus comprised only three species: *M. filicaudatus,*
*M. flagellicaudatus Schuurmans Stekhoven, 1946* and *M. effilatus Schuurmans Stekhoven,*
*1946. Metacylicolaimus effilatus* is characterised by a narrow buccal cavity with
longitudinal cuticularised ridges but apparently lacking teeth; other morphological features
of the species agree well with the definition of the genus. Further work is required to

determine whether *M. effilatus* possesses teeth and confirm its taxonomic status. In my experience of making observations of *M. catherinae* sp. nov. specimens, teeth (even if large) may be completely obscured if the buccal cavity has collapsed, is full of detritus or if the specimen is viewed at the wrong angle. I therefore leave the taxonomic placement of this species unchanged for the time being.

### *Metacylicolaimus catherinae* sp. nov.

Figures 1 and 2, Table 2

urn:lsid:zoobank.org:act:F2B1591E-C796-4333-AA0B-D526F0FBD9B3

**Type locality:** Hikurangi Margin off east coast of New Zealand's North Island, Mungaroa seep site, 2,075–2,076 m water depth, sediment depth 0–5 cm, *R. Tangaroa* voyage TAN1904, stations 19 (41.9378°S, 175.3112°W) and 20 (41.4260°S, 176.3118°W).

**Type material**: Holotype male (NIWA 154925), two paratype males and four paratype females (NIWA 154926-154927), collected in July 2019.

**Measurements:** See Table 2 for detailed measurements.

**Description:** Males. Body large, colourless, tapering slightly towards anterior extremity. Cuticle smooth, with some slight, oblique cross-striations visible in some specimens, 8–11 µm thick throughout body. Eight longitudinal rows of sparsely distributed setae in pharyngeal region and papillae in main body region, each connected to an epidermal gland. Type I loxometanemes (*i.e.*, metanemes running obliquely to longitudinal axis but restricted to lateral chords), with both caudal and frontal filaments. Cephalic capsule present, consisting of a band of thickened endocuticle and with faint posterior outline (suture) slightly posterior to cephalic setae and with invaginating around amphidial fovea. Labial region rounded, consisting of three lips each bearing two inner labial papillae and with inner serrated layer tapering distally. Six outer labial setae and four cephalic setae of same length (0.22–0.30 cbd long) and located in one circle; lateral outer labial setae displaced slightly ventrally relative to amphidial fovea. Group of 2–4 subcephalic setae present posterior to amphids, 8–12 µm long. Amphidial aperture narrow, elliptical, located at anterior edge of pocket-shaped amphidial fovea with slightly cuticularized outline and with proximal portion of amphidial nerve clearly visible. Ocelli absent. Buccal cavity deep, relatively wide and funnel-shaped in some specimens, collapsed and narrow in others (making teeth difficult to observe). Large cuticularized dorsal tooth in anterior half of buccal cavity, followed by smaller, blunt dorsal tooth in posterior half of buccal cavity; relatively large and blunt ventral (possibly ventrosublateral) tooth also present. Pharynx muscular, cylindrical, widening slightly posteriorly; pharyngeal lumen not cuticularised. Pharyngeal glands ducts extending to base of posterior dorsal and ventral teeth. Cardia triangular, ca. 44–52 µm long, surrounded by intestine. Nerve ring surrounding pharynx slightly posterior to one quarter of pharynx length from anterior. Secretory-excretory system not observed.

Reproductive system diorchic with opposed and outstretched testes; anterior testis located to the right or left of intestine, posterior testis on same side or opposite side. Sperm

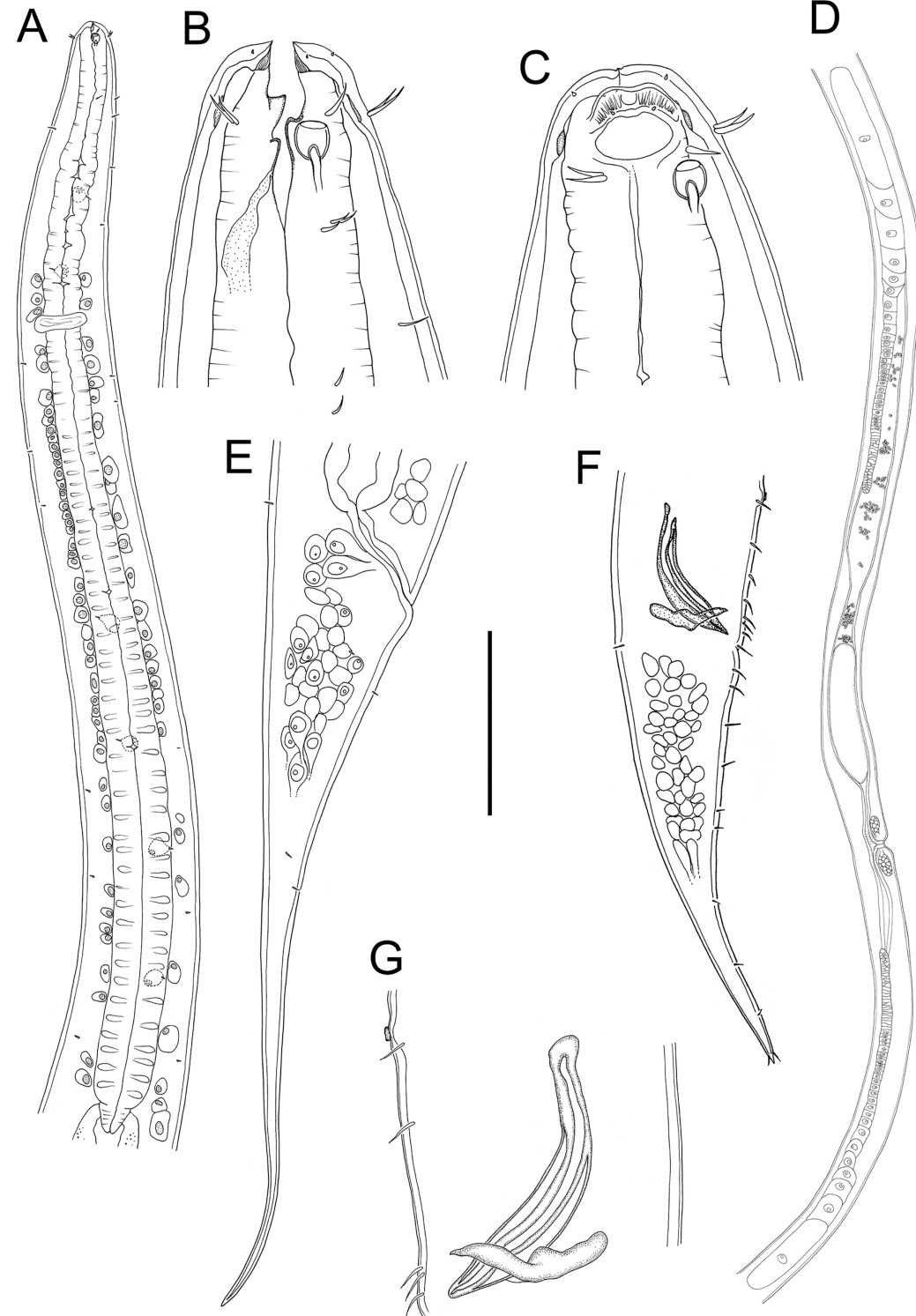

**Figure 1** *Metacylicolaimus catherinae* **sp. nov.** (A) Male pharyngeal body region; (B and C) female cephalic region; (D) female reproductive system; (E) female posterior body region; (F) male posterior body region; (G) male copulatory apparatus. Scale bar: A = 300 μm, B and C = 70 μm, D = 800 μm, E = 175 μm, F = 215 μm, G = 100 μm.

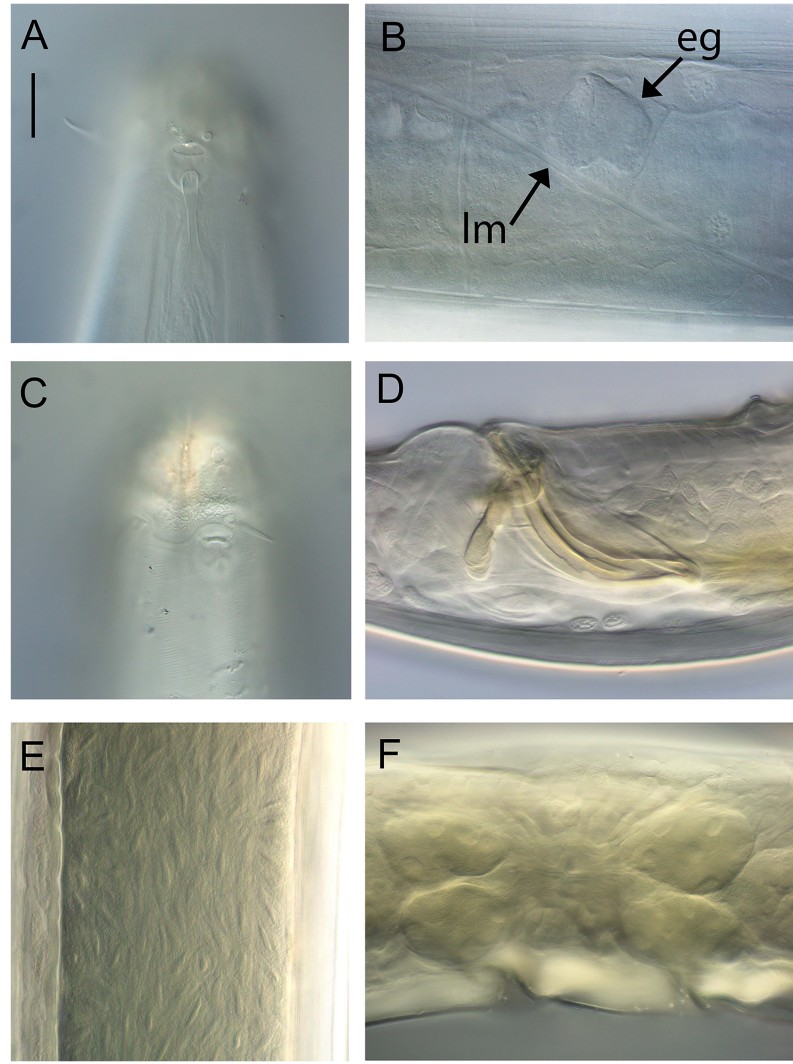

**Figure 2** ***Metacylicolaimus catherinae*** **sp. nov.** Light micrographs. (A and C) Male cephalic region; (B) lateral view of mid-body region showing type I loxometaneme (lm) and epifermal gland (eg); (D) male copulatory apparatus; (E) sperm; (F) ventral view of vulva. Scale bar: A and C = 20 μm, B = 13 μm, D = 16 μm, E = 30 μm, F = 50 μm.

spindle-shaped, ca. 9–12 × 21–32 μm. Spicules paired, symmetrical, curved, 1.2 cloacal body diameter long, widest near proximal end then tapering distally. Gubernaculum with lateral, distally pointed processes flanking the spicules, and strongly cuticularized dorso-caudally, dorsally or dorso-anteriorly directed apophyses. Twelve to twenty-one pericloacal setae present on either side of the cloaca; precloacal seta not observed. One disc-like precloacal supplement present, located 140–172 μm anterior to cloaca. Tail conical, with pointed tail tip; sparse subventral and subdorsal setae present, 6 μm long, two short terminal setae present. Caudal glands and spinneret not observed.

Females. Similar to male, but with longer tail. Reproductive system with two opposed, reflexed ovaries; anterior ovary located to the left of intestine, posterior ovary located to the left or right of intestine. Spermatheca not observed. Vulva situated near mid-body. Vagina

**Table 2 Morphometrics (μm) of _Metacylicolaimus catherinae_ sp. nov.**

|  | Male | | | Females | | | |
|---|---|---|---|---|---|---|---|
|  | Holotype | Paratypes | | Paratypes | | | |
| Specimen | M1 | M2 | M3 | F1 | F2 | F3 | F4 |
| L | 16,525 | 17,144 | 15,514 | 13,454 | 17,062 | 15,721 | 18,943 |
| a | 65 | 75 | 55 | 63 | 59 | 66 | 56 |
| b | 9 | 10 | 9 | 8 | 9 | 8 | 8 |
| c | 26 | 38 | 26 | 22 | 25 | 21 | 23 |
| c′ | 4.3 | 3.2 | 3.8 | 4.5 | 4.8 | 5.6 | 4.7 |
| Head diam. at cephalic setae | 60 | 59 | 57 | 64 | 63 | 61 | 67 |
| Head diam. at amphids | 66 | 61 | 71 | 62 | 66 | 63 | 72 |
| Length of outer labial setae | 15, 16 | 15, 16 | 13–16 | 17 | 18 | 14 | 18 |
| Length of cephalic setae | 16, 17 | 13–17 | 16 | 18 | 19 | 18 | 19 |
| Amphidial fovea height | 16 | 16 | 17 | 18 | 17 | 17 | 22 |
| Amphidial fovea width | 14 | 16 | 15 | 15 | 14 | 15 | 15 |
| Amphidial fovea width/cbd (%) | 24 | 23 | 21 | 24 | 21 | 24 | 21 |
| Amphidial fovea from anterior end | 33 | 28 | 36 | 15 | 28 | 30 | 29 |
| Amphidial aperture height | 8 | 9 | 10 | 9 | 11 | 9 | 11 |
| Amphidial aperture width | 3 | 4 | 3 | 3 | 5 | 4 | 5 |
| Nerve ring from anterior end | 589 | 487 | 510 | 493 | 527 | 515 | 695 |
| Nerve ring cbd | 180 | 159 | 168 | 188 | 177 | 170 | 213 |
| Pharynx length | 1,912 | 1,772 | 1,679 | 1,693 | 1,825 | 1,864 | 2,370 |
| Pharyngeal diam. at base | 111 | 99 | 121 | 127 | 127 | 111 | 149 |
| Pharynx cbd at base | 255 | 221 | 278 | 293 | 290 | 237 | 324 |
| Max. body diam. | 255 | 228 | 280 | 214 | 290 | 237 | 340 |
| Spicule length | 171 | 175 | 189 | – | – | – | – |
| Gubernaculum length | 45 | 43 | 49 | – | – | – | – |
| Cloacal/anal body diam. | 146 | 142 | 158 | 136 | 145 | 134 | 177 |
| Tail length | 634 | 456 | 607 | 610 | 689 | 756 | 838 |
| V | – | – | – | 8,403 | 8,669 | 8,024 | 9,586 |
| %V | – | – | – | 62 | 51 | 51 | 51 |
| Vulval body diam. | – | – | – | 211 | 214 | 215 | 226 |

Note:
a, body length/maximum body diameter; b, body length/pharynx length; c, body length/tail length; c′, tail length/anal or cloacal body diameter; cbd, corresponding body diameter; L, total body length; V, vulva distance from anterior end of body; %V, V/total body length × 100.

perpendicular, short, without cuticularisation; two pair of large vaginal glands located anterior and posterior to vulva.

**Diagnosis:** _Metacylicolaimus catherinae_ sp. nov. is characterised by body length 13.5–18.9 mm; three lips with inner serrated layer tapering distally; inner labial setae and cephalic setae 0.22–0.30 cbd long; buccal cavity with anterior and posterior dorsal teeth and single ventral (possibly ventrosublateral) tooth; nerve ring slightly posterior to one quarter of pharynx length from anterior; spicules 1.2 cloacal body diameter long, gubernaculum with

lateral, distally pointed processes flanking the spicules and dorsally directed apophyses rounded distally; 12–21 pericloacal setae present on either side of the cloaca and one disc-like precloacal supplement present; tail conical, with pointed tail tip, 3.2–4.3 cloacal body diameters long in males and 4.5–5.6 canal body diameters in females.

**Differential diagnosis:** The new species differs from *M. filicaudatus* in body length (13.5–18.9 *vs.* 12.3–13.0 mm in *M. filicaudatus*), ratio of a (55–77 *vs.* 37–39 in *M. filicaudatus*), b (8–10 *vs.* 5–7 in *M. filicaudatus*), and c (21–38 *vs.* 18–19 in *M. filicaudatus*), buccal cavity armature (two dorsal teeth and one ventral tooth *vs.* single dorsal tooth in *M. filicaudatus*), precloacal supplement structure (disc-like *vs.* tubular in *M. filicaudatus*) and tail shape (conical with pointed tip *vs.* conicocylindrical with rounded tip in *M. filicaudatus*). *Metacylicolaimus catherinae* sp. nov. differs from *M. flagellicaudatus* in body length (13.5–18.9 *vs.* 7.7–11.6 mm in *M. flagellicaudatus*), ratio of a (55–77 *vs.* 32–44 in *M. flagellicaudatus*) and c (21–38 *vs.* 11–15 in *M. flagellicaudatus*), buccal cavity armature (anterior and posterior dorsal teeth and single ventral tooth *vs.* dorsal tooth only in *M. flagellicaudatus*), tail shape (conical and gradually tapering *vs.* conicocylindrical with filiform posterior portion in *M. flagellicaudatus*) and longer tail (5.2–6.0 *vs.* 4.3 cloacal/anal body diameters in *M. flagellicaudatus*). Finally, *M. catherinae* sp. nov. differs from *M. effilatus* in the ratios of a (55–77 *vs.* 49–52 in *M. effilatus*), b (8–10 *vs.* 3–6 in *M. effilatus*), and c (21–38 *vs.* 16 in *M. effilatus*), buccal cavity armature (anterior and posterior dorsal teeth and single ventral tooth *vs.* teeth apparently absent in *M. effilatus*), and wider amphids (35–43 *vs.* 19% cbd wide in *M. effilatus*).

**Etymology:** The species is named after Catherine Leduc, the author's sister.

Family Oncholaimidae *Filipjev, 1916*

**Family diagnosis:** (from *Smol, Muthumbi & Sharma (2014)*) Lips usually merged. Buccal cavity spacious, posteriorly surrounded by pharyngeal tissue, usually with three unequal immovable teeth. One of the two ventrosublateral teeth usually the largest; sometimes the two ventrosublateral teeth equal in size and bigger than the dorsal tooth, rarely all three teeth equal in size. Pharynx not inserted into the body wall. Outline of pharynx smooth. Three pharyngeal glands open through teeth. Amphids generally pocket-shaped (dorsally spiral in many Enchelidiidae). Delicately built dorsolateral and ventrolateral orthometanemes with pronounced caudal filament. The secretory-excretory system distinct with gland cell to right side of intestine. Female reproductive system variable: either didelphic-amphidelphic or monodelphic-prodelphic. The demanian system absent or present in different degrees of development. Males mostly diorchic with opposed testes. Spicules of variable shape, gubernaculum present or absent. Gonads usually on right side of intestine.

Subfamily Oncholaiminae *Filipjev, 1916*

**Subfamily diagnosis:** (from *Smol, Muthumbi & Sharma (2014)*) Left ventrosublateral tooth almost always larger than the other teeth; very rarely all three teeth of same size.

Females monodelphic-prodelphic with antidromously reflexed ovary. Demanian system present or absent.

Genus *Oncholaimus* Dujardin, 1845

**Genus diagnosis: (from** Smol, Muthumbi & Sharma (2014)**)** Left ventrosublateral tooth almost always largest. Female reproductive system with demanian system, terminal ducts and pores present in variable number or absent in virgin females. Males diorchic. Spicules short, gubernaculum absent. Tail short.

= *Oncholaimium* Cobb, 1930

Type species: *O. attenuatus* Dujardin, 1845

**Remarks.** Shimada et al. (2017) provided a key to species of *Oncholaimus* with a conicocylindrical tail.

***Oncholaimus adustus* sp. nov.**
Figures 3–5, Table 3
urn:lsid:zoobank.org:act:4019665C-3216-4225-903B-93D61916F5B1

**Type locality:** Hikurangi Margin off east coast of New Zealand's North Island, Mungaroa and Uruti South seep sites, 2,077 and 1,232–1,245 m water depth, respectively, sediment depth 0–5 cm, *R. Tangaroa* voyage TAN1904, stations 21 (41.9382°S, 175.3128°W), 64 (41.4279°S, 176.3485°W), 67 (41.4264°S, 176.3500°W) and 68 (41.4260°S, 176.3506°W).

**Type material**: Holotype male (NIWA 154928), two paratype males and four paratype females (NIWA 154929–154931), collected in July 2019.

**Measurements:** See Table 3 for detailed measurements.

**Description:** Males. Body colourless, except for intestine wall (and to a lesser extent pharynx) which has brown colouration in most specimens; body tapering slightly near both extremities. Cuticle smooth. Eight longitudinal rows of sparsely distributed somatic setae; epidermal glands not observed. Orthometanemes present, with frontal filaments sometimes overlapping; caudal filaments apparently absent. Labial region with six lips tapering distally, each bearing a small labial papilla. Outer labial setae in same circle as the cephalic setae; outer labial setae ca. 0.2 cbd long, slightly shorter than cephalic setae ca. 0.3 cbd long. Amphidial fovea pocket-shaped and elliptical amphidial aperture located near middle of buccal cavity, 36–43% cbd wide. Ocelli absent. Buccal cavity large, barrel-shaped, with strongly cuticularized walls, 74–78 μm deep and up to 26–33 μm wide. Three large, blunt teeth either equal in size or with left ventrosublateral tooth appearing slightly larger than right ventrosublateral and dorsal teeth; all teeth reaching to about 3/4 of buccal cavity length. Pharynx muscular, cylindrical, widening posteriorly without forming bulb; pharyngeal lumen not cuticularised. Pharyngeal glands not observed; pharyngeal ducts visible anteriorly, extending into and opening near tip of each tooth. Cardia well-developed, ca. 44–58 μm long, triangular in shape, almost completely surrounded by intestine tissue. Nerve ring surrounding pharynx slightly anterior to middle of pharynx length. Secretory-excretory system present; pore located slightly posterior to buccal cavity,

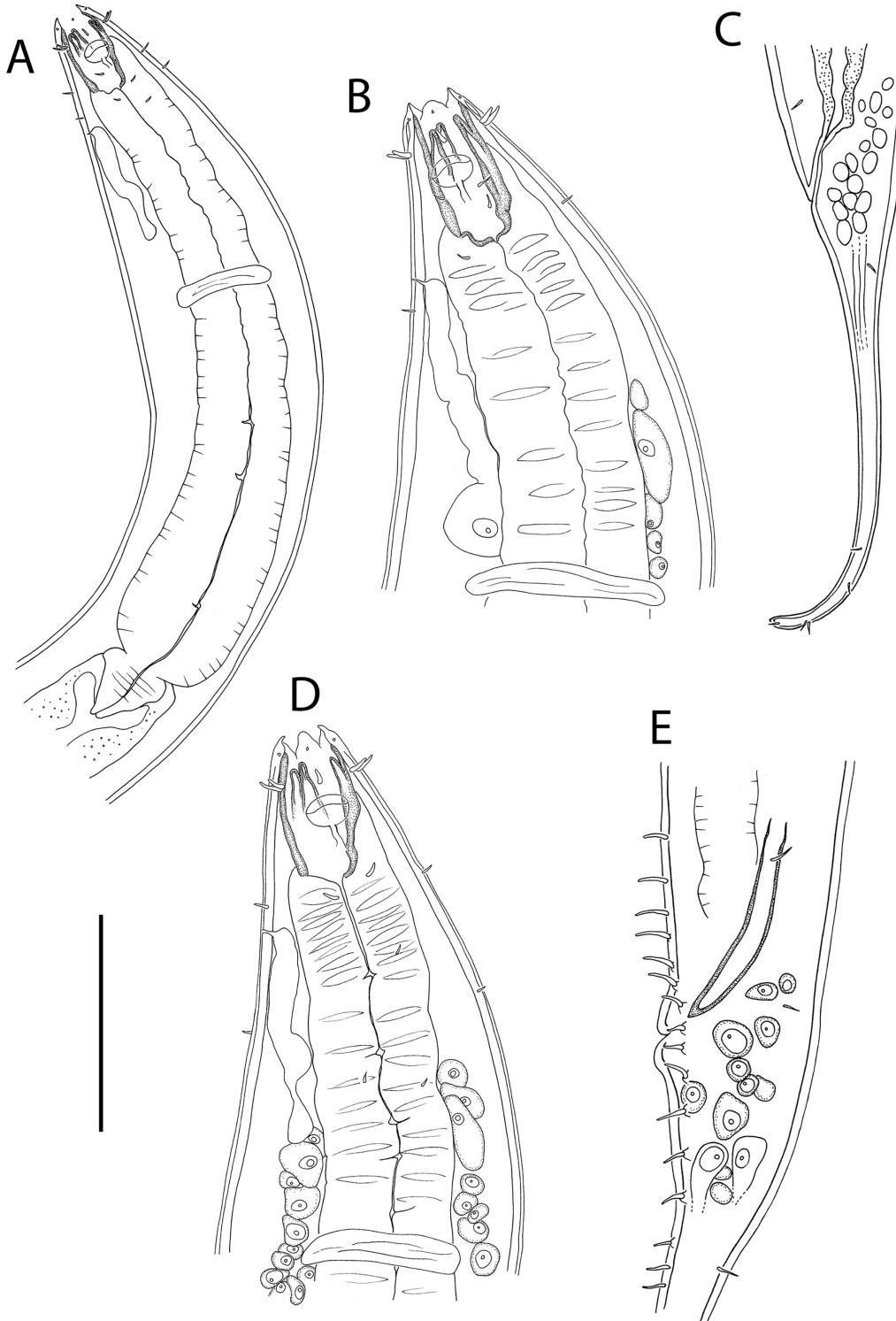

**Figure 3** *Oncholaimus adustus* **sp. nov.** (A) Male pharyngeal body region; (B) female cephalic region; (C) female posterior body region; (D) female cephalic region; (E) male copulatory apparatus. Scale bar: A = 200 μm, B and D = 116 μm, C = 130 μm, E = 70 μm.

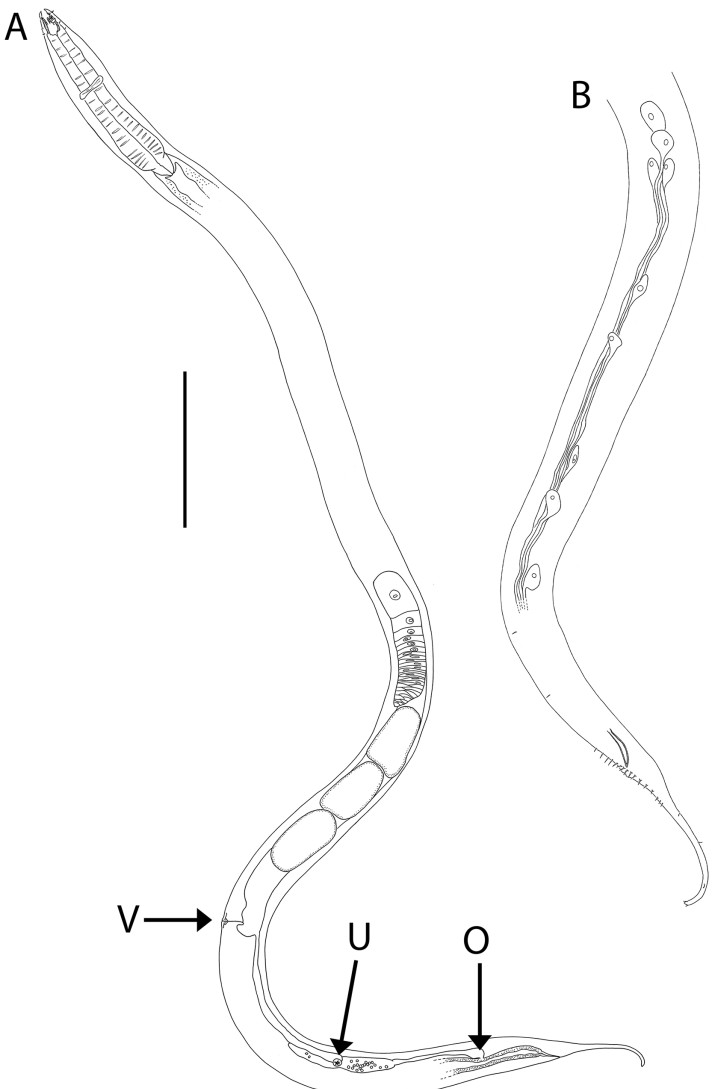

**Figure 4** *Oncholaimus adustus* **sp. nov.** (A) Entire female showing location of vulva (V), uvette (U) and osmium (O); (B) male posterior body region showing caudal glands. Scale bar: A = 500 µm, B = 250 µm.

ca. 0.90–0.95 cbd from anterior body extremity, ventral gland small and located anterior to nerve ring.

Reproductive system diorchic with opposed and outstretched testes; anterior testis located to the right of intestine, posterior testis to the left of intestine. Sperm globular, ca. 11–12 × 20 µm. Spicules paired, symmetrical, slightly bent and tapering distally, 1.4 cloacal body diameter long. Gubernaculum absent. Precloacal sensillum absent. Twelve to sixteen circum-cloacal setae on each ventrolateral sides, 7–13 µm long. Eight to ten ejaculatory glands present laterally on either side of intestine, extending up to ca. 1,000 µm anterior to cloaca. Tail conicocylindrical, with sparse sublateral and subdorsal setae, 6 µm long, tail tip not swollen. Caudal glands and numerous pseudocoelomocytes present; spinneret not observed.

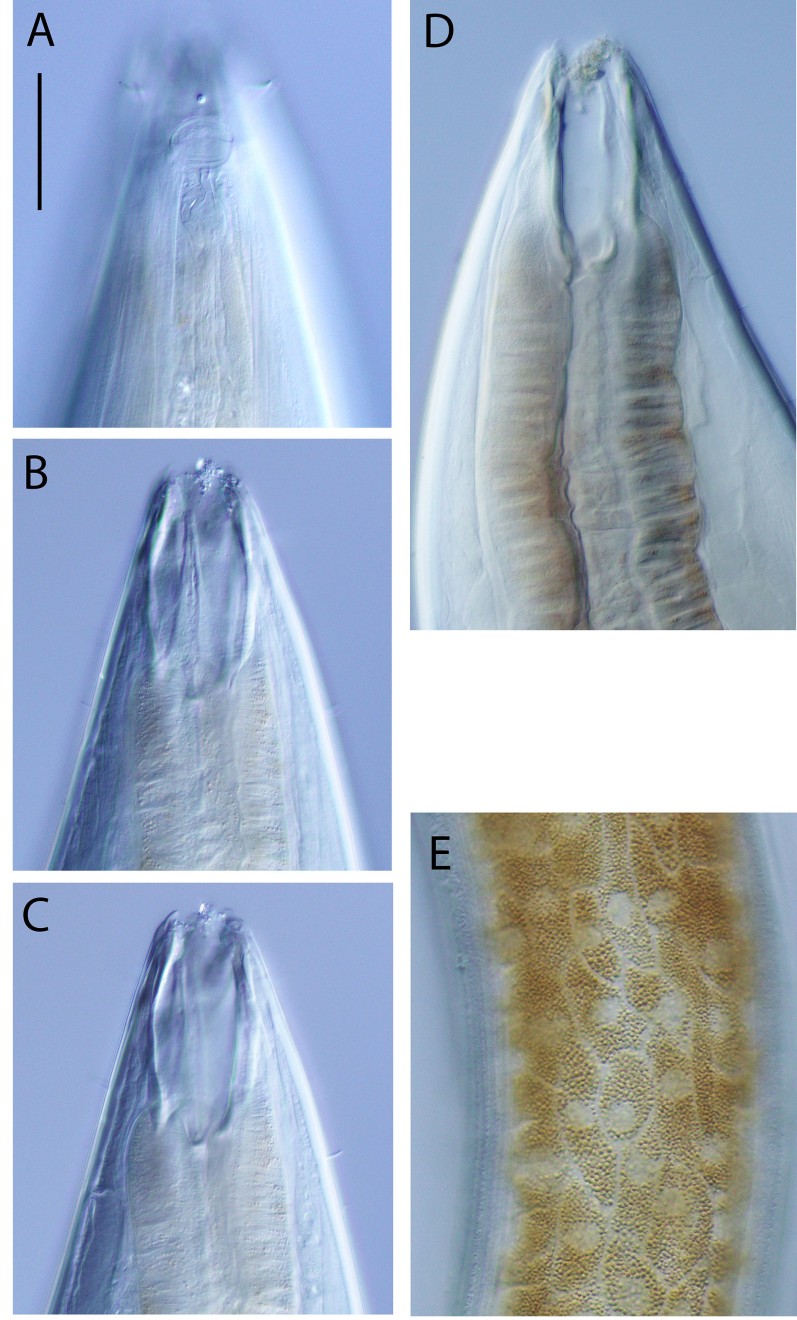

**Figure 5 *Oncholaimus adustus* sp. nov.** Light micrographs. (A–C) Male cephalic region showing amphid, anterior sensilla, buccal cavity and armature, and secretory-excretory pore; (D) female anterior body region; (E) intestine wall. Scale bar: A–C = 50 µm, D = 45 µm, E = 55 µm.

Females. Similar to males. Reproductive system with single anterior reflexed ovary located to the right or left of intestine; eggs ca. 180–280 × 95 µm. Vulva situated near two thirds of body length from anterior extremity. Vagina perpendicular, without cuticularisation; vaginal glands not observed. Demanian system present. Ductus uterinus

**Table 3 Morphometrics (μm) of *Oncholaimus adustus* sp. nov.**

| | Males | | | Females | | | |
|---|---|---|---|---|---|---|---|
| | Holotype | Paratypes | | Paratypes | | | |
| Specimen | M1 | M2 | M3 | F1 | F2 | F3 | F4 |
| L | 7,427 | 6,319 | 6,472 | 6,568 | 4,318 | 7,692 | 4,472 |
| a | 45 | 46 | 45 | 35 | 35 | 44 | 36 |
| b | 13 | 12 | 12 | 11 | 9 | 13 | 9 |
| c | 25 | 22 | 25 | 18 | 17 | 22 | 16 |
| c′ | 5.8 | 5.4 | 5.5 | 6.0 | 5.5 | 5.4 | 5.2 |
| Head diam. at cephalic setae | 50 | 47 | 44 | 52 | 40 | 45 | 40 |
| Head diam. at amphids | 61 | 56 | 54 | 65 | 52 | 59 | 54 |
| Length of outer labial setae | 10–11 | 10 | 9–10 | 11–12 | 9–10 | 8 | 8–9 |
| Length of cephalic setae | 12–15 | 13–14 | 12–13 | 13–15 | 10–11 | 12 | 10–11 |
| Amphidial aperture height | 5 | 6 | 5 | 6 | 5 | 5 | 5 |
| Amphidial fovea height | 16 | 17 | 16 | 17 | 15 | 17 | 15 |
| Amphidial aperture width | 23 | 20 | 23 | 23 | 19 | 24 | 19 |
| Amphidial aperture width/cbd (%) | 38 | 36 | 43 | 35 | 37 | 41 | 35 |
| Amphidial aperture from anterior end | 34 | 32 | 27 | 33 | 30 | 27 | 27 |
| Nerve ring from anterior end | 272 | 271 | 247 | 282 | 238 | 290 | 237 |
| Nerve ring cbd | 165 | 137 | 143 | 190 | 123 | 172 | 123 |
| Pharynx length | 572 | 529 | 522 | 611 | 465 | 605 | 480 |
| Pharyngeal diam. at base | 111 | 87 | 89 | 135 | 95 | 114 | 87 |
| Pharynx cbd at base | 152 | 123 | 134 | 178 | 117 | 161 | 118 |
| Max. body diam. | 165 | 137 | 143 | 190 | 124 | 172 | 125 |
| Spicule length | 75 | 73 | 65 | – | – | – | – |
| Cloacal/anal body diam. | 52 | 52 | 48 | 63 | 45 | 65 | 53 |
| Tail length | 299 | 282 | 263 | 375 | 247 | 351 | 278 |
| V | – | – | – | 4,889 | 2,917 | 5,505 | 3,152 |
| %V | – | – | – | 74 | 68 | 72 | 70 |
| Vulval body diam. | – | – | – | 155 | 124 | 153 | 112 |

Note:
a, body length/maximum body diameter; b, body length/pharynx length; c, body length/tail length; c′, tail length/anal or cloacal body diameter; cbd, corresponding body diameter; L, total body length; V, vulva distance from anterior end of body; %V, V/total body length × 100.

extending posteriorly from uterus with uvette located 130–1,040 μm posterior to vulva; ductus entericus posterior to uvette, connecting to intestine *via* osmium ca. 75–365 μm anterior to anus. Terminal ducts and pores not observed.

**Diagnosis:** *Oncholaimus adustus* sp. nov. is characterised by body length 6,319–7,427 μm in males and 4,318–7,692 μm in females; outer labial setae ca. 0.2 cbd long, cephalic setae ca. 0.3 cbd long; amphidial aperture located near middle of buccal cavity and ca. 0.4 cbd wide; buccal cavity with three equal teeth or with left ventrosublateral tooth slightly larger than other two teeth, all reaching to about 3/4 of buccal cavity length; secretory-excretory pore located near or slightly posterior to posterior edge of buccal cavity, ventral

gland located anterior to nerve ring; spicules 1.4 cloacal body diameter long, 12–16 circum-cloacal setae on each ventrolateral sides; demanian system with uvette located 130–1,040 µm posterior to vulva and osmium located posterior to uvette; conicocylindrical tail 5.2–6.0 cloacal/anal body diameters long.

**Differential diagnosis:** The new species is most similar to *O. problematicus Coles, 1977* (described based on specimens from South Africa coast at 50–183 m depth) in body length of males (>6 mm), amphids near middle of buccal cavity, secretory-excretory pore posterior to buccal cavity (in at least some specimens), spicules <2 cloacal body diameters long and shorter than tail, and demanian system present. One of the females described by *Coles (1977)* also has equal teeth, an unusual trait for the genus which is also found in *O. adustus* sp. nov. The new species differs from *O. problematicus* in having a shorter pharynx (465–611 *vs.* 770–950 µm in *O. problematicus*), secretory-excretory pore located further anteriorly (86–102 *vs.* 110–150 µm from anterior body extremity in *O. problematicus*), shorter spicules (65–75 *vs.* 90–120 µm in *O. problematicus*) and larger amphids (width 19–24 µm or 35–43% cbd *vs.* 10–15 µm or 29–30% cbd in *O. problematicus*).

The new species is similar to *O. ramosus Smolanko & Belogurov, 1987* in having a long body >6 mm, secretory-excretory pore located at posterior end of buccal cavity (in some specimens), spicules <2 cloacal body diameters long and shorter than tail, and presence of a demanian system. *Oncholaimus adustus* sp. nov. can be differentiated from the latter by the equal or slightly unequal teeth all reaching to about 3/4 of buccal cavity length from posterior (*vs.* left ventrosublateral tooth clearly larger than the other two teeth which reach to half of buccal cavity length in *O. ramosus*), six conical lips tapering distally (*vs.* six blunt lips in *O. ramosus*), and osmium located far posteriorly close to anus (*vs.* osmium only slightly posterior to uvette and far anteriorly relative to anus in *O. ramosus*).

*Oncholaimus adustus* sp. nov. is also similar to *O. keiensis Kreis, 1932* in having equal or nearly equal teeth, but can be differentiated from the latter by the substantially longer body (females 4.3–7.7 *vs.* 2.4–2.5 µm in *O. keiensis*), teeth reaching to about 3/4 of buccal cavity length from posterior (*vs.* teeth reaching to half of buccal cavity length in *O. keiensis*), and vulva located further posteriorly (68–74 *vs.* 45% of body length from anterior extremity in *O. keiensis*).

**Etymology:** The species name is derived from the Latin *adustus* (= tanned, brown, swarthy), which refers to the conspicuous colour of the intestine wall of most specimens.
Family Oxystominidae *Chitwood, 1935*

**Family diagnosis: (From *Smol, Muthumbi & Sharma (2014)*)** Body elongated and very thin at the anterior end. Anterior sensilla in three separate circles, the second and third circles clearly separated; the inner labial sensilla papilliform or setiform, outer labial and cephalic setae. Buccal cavity narrow, tubular or funnel-shaped and without teeth. Only orthometanemes with very short caudal filaments present. Pharynx inserts into the body cuticle in the region of the buccal cavity, however the cephalic capsule is not well developed. The posterior section of the pharynx has an undulating outline. Females

didelphic-amphidelphic with antidromously reflexed ovaries or monodelphic-opisthodelphic. Males diorchic with opposed testes or only one anterior testis. Position of caudal glands variable.

Subfamily Halalaiminae *De Coninck, 1965*

**Subfamily diagnosis: (from** *Smol, Muthumbi & Sharma (2014)*) Amphidial aperture a longitudinal groove, fovea nearly non-existent.

Genus *Halalaimus de Man, 1888*

**Generic diagnosis: (from** *Smol, Muthumbi & Sharma (2014)*) Anterior and posterior end of body strongly attenuated. Amphid greatly elongated longitudinally. Anterior sensilla in three circles: six inner labial sensilla sometimes indistinct, papilliform or setiform; six outer labial setae and four cephalic setae. Cuticle thin from the anterior end to the level of the cephalic setae and thickened posterior to it; fine transverse striations present or absent. Alae-like structures at lateral chords present in some species. Buccal cavity absent. Pharynx long, narrow anteriorly and broader posteriorly. Female reproductive system didelphic-amphidelphic with reflexed ovaries. Males diorchic with opposed testes; precloacal sensillum (seta) and/or pore present or absent; caudal alae present or absent. Egg and sperm dimorphism occurs in at least one species. Tail conicocylindrical, tip blunt or bifurcate, cylindrical part of the tail with or without transverse cuticular striations. Caudal glands present, spinneret present or undetermined.

Type species: *H. gracilis de Man, 1888*

**Remarks.** A revision of the genus and key to species were provided by *Keppner (1992)*. The latter author divided the genus into four groups of species based on male characters including presence/absence of caudal alae, and presence/absence of a precloacal sensillum and/or pore. Additional *Halalaimus* species were subsequently described by *Bussau (1993)*, *Alekseev & Linnik (1994)*, *Turpeenniemi (1997)*, *Pastor de Ward (1998)*, *Nasira & Turpeenniemi (2002)*, *Gagarin & Thanh (2004*, *2014*, *2018)*, *Huang & Zhang (2005)*, *Gagarin (2016*, *2020)* and *Shimada et al. (2020)*. *Shimada et al. (2020)* provided an updated key to *Halalaimus* species of group 2, which consists of species characterised by males with caudal alae and lacking a precloacal sensillum or pore.

### *Halalaimuss talaurinus* sp. nov.

Figures 6–8, Table 4

urn:lsid:zoobank.org:act:DE927E2B-B780-40DB-AE04-4EE038F5AFF1

**Type locality:** Hikurangi Margin off east coast of New Zealand's North Island, Uruti South seep site, 1,230–1,245 m water depth, sediment depth 0–5 cm, *R. Tangaroa* voyage TAN1904, stations 64 (41.4279°S, 176.3485°W), 66 (41.4266°S, 176.3497°W) and 67 (41.4264°S, 176.3500°W).

**Type material**: Holotype male (NIWA 154932), one paratype male and three paratype females (NIWA 154933–154934), collected in July 2019.

**Measurements:** See Table 4 for detailed measurements.

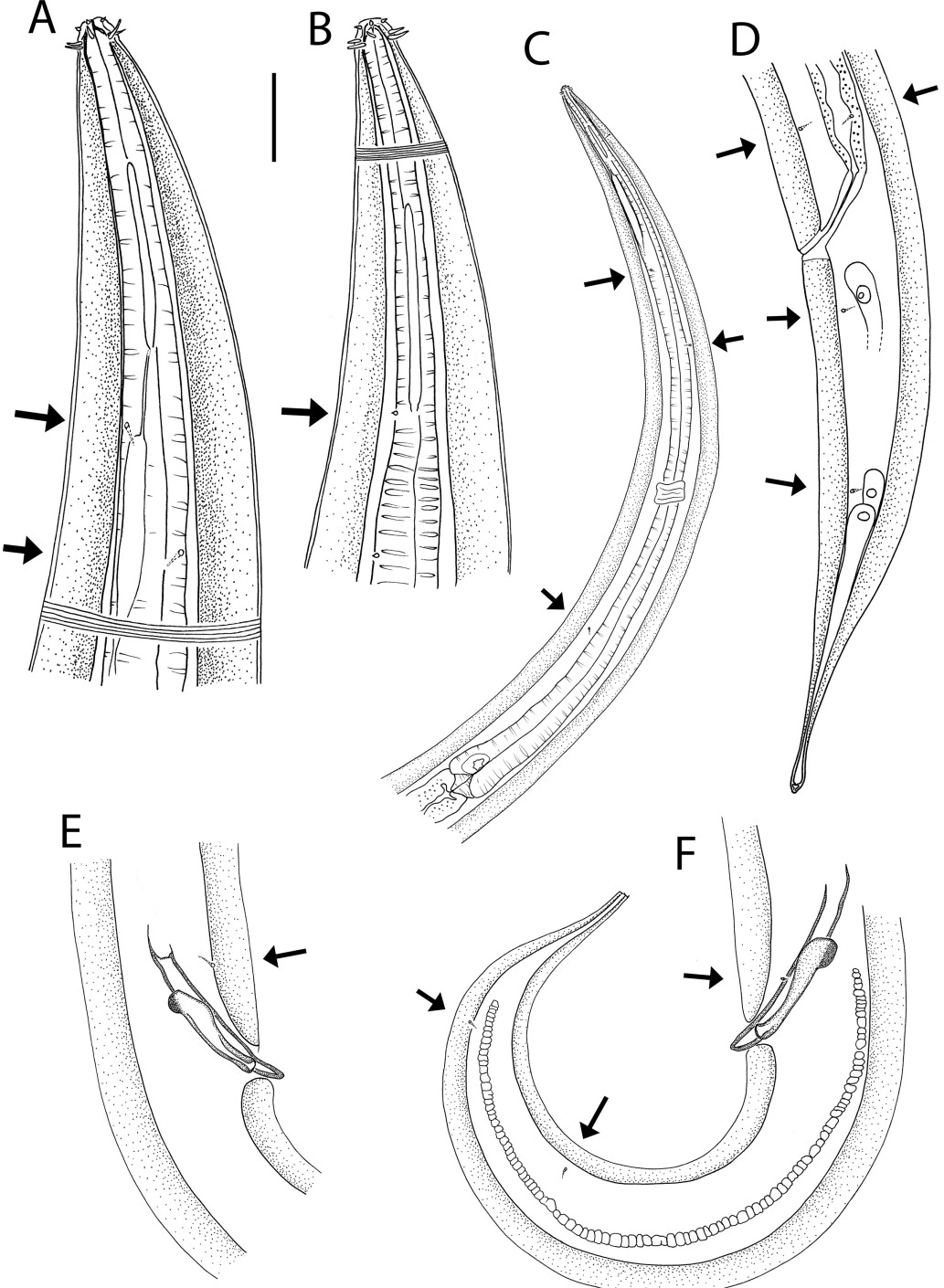

**Figure 6** *Halalaimus talaurinus* **sp. nov.** (A) Female anterior body region; (B) male anterior body region; (C) male pharyngeal body region; (D) posterior body region; (E) male copulatory apparatus; (F) male posterior body region (tail tip broken). Arrows show position of cuticle pores. Scale bar: A and B = 25 μm, C = 100 μm, D = 62 μm, E and F = 55 μm.

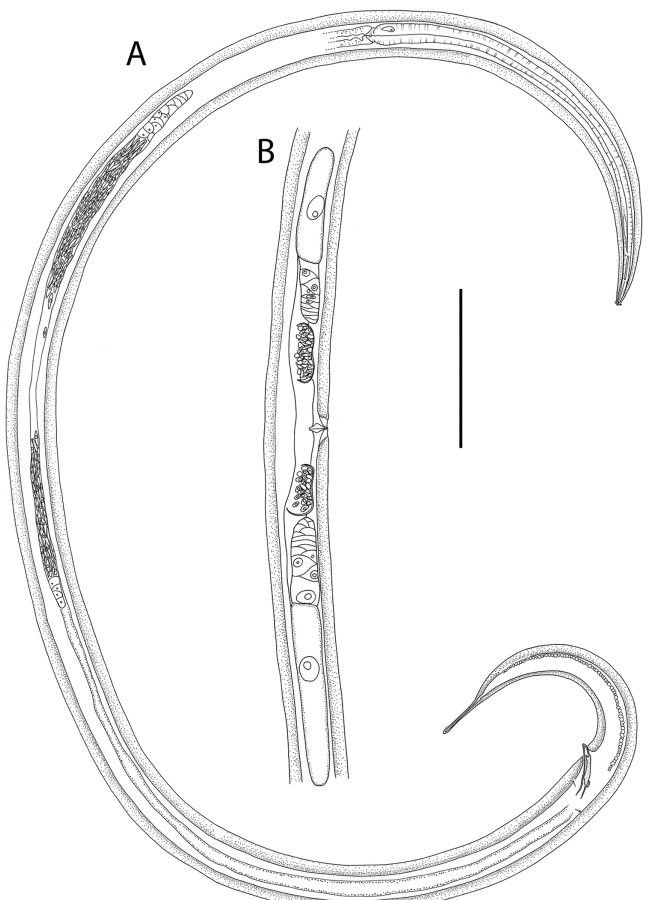

**Figure 7 *Halalaimus talaurinus* sp. nov.** (A) Entire male; (B) female reproductive system. Scale bar = 200 μm.

**Description:** Males. Body colourless, tapering towards anterior extremity, curved ventrally. Cuticle with slight transverse striations beginning between cephalic setae and amphids, without lateral differentiation except in caudal region where ornamented (*i.e.*, with scale-like structures) lateral alae extend from level of cloaca to about three quarters of tail length. Cuticle in cephalic region thickening posterior to cephalic setae to become extremely thick at level of amphid and throughout rest of body except near tail tip where it becomes gradually thinner again; cuticle thickness 14–16 μm at base of pharynx, 16–18 μm in mid-body and 15–17 μm in precloacal region, with cuticle thickness equivalent to 40–52% of body radius in widest part of body. Eight longitudinal rows of sparsely distributed cuticle pores distributed along almost entire body length; epidermal glands not observed. Orthometanemes present, apparently few in number; caudal and frontal filaments indistinct. Labial region slightly rounded, slightly offset from body contour. Six inner labial papillae ca. 1 mm long located in separate circle from six outer labial setae and four cephalic setae; outer labial setae slightly shorter and located immediately anterior to the four cephalic setae, outer labial setae 0.25–0.43 cbd long, cephalic setae 0.38–0.50 cbd long. Amphidial aperture longitudinally elongated, ca. 23 times as high as

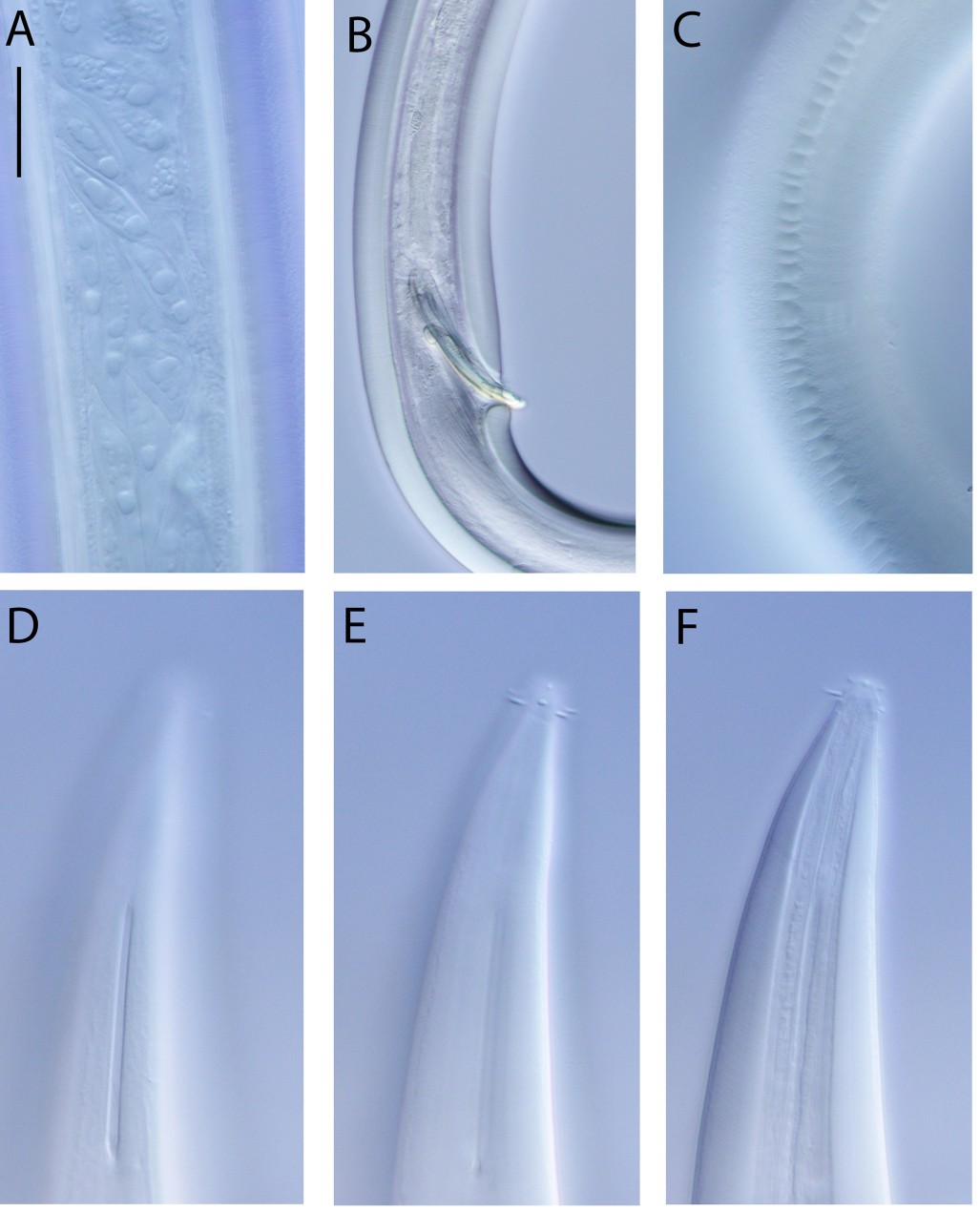

**Figure 8 *Halalaimus talaurinus* sp. nov.** Light micrographs. (A) Male testis showing sperm; (B) male copulatory apparatus; (C) male caudal ala; (D–F) male cephalic region showing amphidial aperture, anterior sensilla and cuticle thickness. Scale bar: A–C = 20 mm; D–F = 15 μm.

wide. Ocelli absent. Buccal cavity minute. Pharynx muscular, cylindrical, widening posteriorly; pharyngeal lumen not cuticularised. Pharyngeal glands visible, with ducts extending to level of amphidial aperture where they may connect to pharyngeal lumen or buccal cavity however their orifices indistinct. Cardia well-developed, ca. 17 μm long, triangular in shape, partially encased in pharyngeal tissue anteriorly and surrounded by

**Table 4 Morphometrics (μm) of *Halalaimus talaurinus* sp. nov.**

| | Males | | Females | | |
|---|---|---|---|---|---|
| | Holotype | Paratype | Paratypes | | |
| Specimen | M1 | M2 | F1 | F2 | F3 |
| L | 2,910 | 2,880 | 2,906 | 2,768 | 2,843 |
| a | 43 | 46 | 34 | 38 | 38 |
| b | 5 | 5 | 5 | 5 | 5 |
| c | 8 | 9* | 11 | 12 | 11 |
| c′ | 5.7 | 5.2* | 5.1 | 4.4 | 4.4 |
| Head diam. at cephalic setae | 7 | 7 | 8 | 8 | 7 |
| Head diam. at amphids** | 24 | 23 | 23 | 25 | 24 |
| Length of outer labial setae | 3 | 3 | 2 | 3 | 3 |
| Length of cephalic setae | 4 | 4 | 3–4 | 4 | 4 |
| Amphidial aperture height | 36 | 34 | 35 | 31 | 31 |
| Amphidial aperture width | 1.6 | 1.5 | 1.5 | 1.8 | 1.4 |
| Amphidial aperture width/cbd (%) | 7 | 7 | 7 | 7 | 6 |
| Amphidial aperture from anterior end | 34 | 26 | 28 | 25 | 25 |
| Nerve ring from anterior end | 322 | 304 | 312 | 295 | 289 |
| Nerve ring cbd | 58 | 58 | 57 | 59 | 55 |
| Pharynx length | 572 | 586 | 571 | 541 | 576 |
| Pharyngeal diam. at base | 29 | 27 | 33 | 29 | 24 |
| Pharynx cbd at base | 66 | 62 | 70 | 64 | 62 |
| Max. body diam. | 68 | 62 | 85 | 73 | 74 |
| Spicule length | 84 | 78 | – | – | – |
| Gubernaculum length | 45 | 50 | – | – | – |
| Cloacal/anal body diam. | 64 | 63 | 54 | 55 | 58 |
| Tail length | 366 | 327* | 275 | 240 | 258 |
| V | – | – | 1,548 | 1,453 | 1,464 |
| %V | – | – | 53 | 52 | 51 |
| Vulval body diam. | – | – | 85 | 73 | 74 |

**Notes:**
* Tail tip missing.
** At middle of amphid.
a, body length/maximum body diameter; b, body length/pharynx length; c, body length/tail length; c′, tail length/anal or cloacal body diameter; cbd, corresponding body diameter; L, total body length; V, vulva distance from anterior end of body; %V, V/total body length × 100.

intestine tissue posteriorly. Nerve ring surrounding pharynx slightly posterior to middle of pharynx length. Secretory-excretory system not observed.

Reproductive system diorchic with opposed and outstretched testes; anterior testis located to the right or left of intestine, posterior testis on same side as anterior testis or shifted slightly ventrally. Sperm spindle-shaped, with 1–3 globular inclusions, ca. 5–6 × 29–34 μm. Spicules paired, symmetrical, almost straight, 1.2–1.3 cloacal body diameter long, tapering distally. Gubernaculum slightly bent, partially encasing spicules laterally, with small, strongly cuticularized dorsocaudal swelling proximally and lateral folds

distally. Precloacal sensillum or pore absent. Tail conicocylindrical, cylindrical portion short; tail tip rounded, slightly swollen. Caudal glands and spinneret not observed.

Females. Similar to male, but with lower values of a and higher values of c. Reproductive system with two opposed, reflexed ovaries; anterior ovary located to the right or left of intestine, posterior ovary located on same side or opposite side. Spermatheca present proximally on each genital branch. Vulva situated near mid-body. Vagina perpendicular, short, without cuticularisation; vaginal glands not observed. Three caudal glands present, spinneret not observed.

**Diagnosis:** *Halalaimus talaurinus* sp. nov. is characterised by body length 2,768–2,910 μm; cuticle 14–18 μm thick (40–52% of body radius at mid-body) between cephalic and caudal regions, lightly striated and without lateral alae except in males which possess ornamented caudal alae; cephalic region slightly offset by constriction; amphidial aperture 31–36 μm high, 1.4–1.8 μm wide; inner labial sensilla papilliform; outer labial setae 0.25–0.43 cbd long, located immediately adjacent to cephalic setae, 0.38–0.50 cbd long; spicules almost straight, 1.2–1.3 cloacal body diameters long, gubernaculum with small, strongly cuticularized dorsocaudal swelling proximally and lateral folds distally, precloacal sensillum or pore absent; conicocylindrical tail 4.4–5.7 cloacal/anal body diameters long, with short cylindrical portion and slightly swollen and rounded posterior extremity.

**Differential diagnosis:** *Halalaimus talaurinus* sp. nov. can be differentiated from all other species of the genus by the extremely thick cuticle (14–18 μm), which makes up 40–52% of the body radius at the body's widest point. *Halalaimus pachyderma* (*Filipjev, 1927*) *Schuurmans Stekhoven, 1935* and *Halalaimus pachydermatus* (*Cobb, 1920*) *Schneider, 1939* are species described as having thick cuticle, however cuticle thickness in *H. pachyderma* does not exceed 7 μm (or 26% of body radius at widest point) whilst the cuticle of *H. pachydermatus* occupies one sixth (17%) of the body radius at its widest point.

The new species belongs to *Keppner (1992)* group 2 of *Halalaimus* species, which are characterised by caudal alae in males (ornamented or unornamented) whilst lacking a precloacal sensillum or pore. Within this group, *H. talaurinus* sp. nov. is the only species characterised by having both ornamented caudal alae and a relatively short tail (<8 cloacal/anal body diameters) with a non-bifurcated tail tip.

**Etymology:** The species name is derived from the Greek *talaurinos* (= shield of tough hide), which refers to the exceptionally thick cuticle of this species.

**Key to valid species of *Halalaimus* (group 2) (updated from *Shimada et al. (2020)*)**

1. Cuticle 14–18 mm thick, comprising 40–2% of body radius at body's widest
   point ................................................................ *H. talaurinus* sp. nov.

- Cuticle comprising <30% of body radius at body's widest point...................... 2

2. Caudal alae unornamented...................................................... 3

- Caudal alae ornamented ........................................................ 9

3. Gubernaculum with dorsocaudally directed apophysis............................... 4

Subfamily Oxystominae *Chitwood, 1935*

**Subfamily diagnosis: (from *Smol, Muthumbi & Sharma (2014)*)** Only dorsolateral orthometanemes. Ventral gland present and confined within the pharyngeal region. Females monodelphic-opisthodelphic. Caudal glands may extend into the precaudal region.

Genus *Thalassoalaimus de Man, 1893*

**Generic diagnosis: (from *Smol, Muthumbi & Sharma (2014)*)** Inner and outer labial sensilla setiform, four cephalic setae in backward position. Amphidial aperture a transverse slit, fovea large and pocket-shaped. Males usually have precloacal papilliform supplements. Thick cuticular lining at the tail tip, called the caudal capsule.

Type species: *T. tardus de Man, 1893*

**Remarks.** The genus was revised by *Martelli et al. (2017)* who provided a key to the identification of all eight valid species based on features of males. There has been some confusion regarding the status of *T. septentrionalis Filipjev, 1927*, *T. septentrionalis spissus Allgén, 1932*, and *T. spissus* (*Allgén, 1932*) *Wieser, 1953*. *Thalassoalaimus septentrionalis spissus* was described by *Allgén (1932)* based on specimens from the coast of Campbell Island. It was considered a subspecies of *T. septentrionalis* (described from the Swedish
coast) by Allgén based on a relatively minor morphological difference, *i.e.*, a more pointed tail tip in *T. septentrionalis spissus* with higher degree of ventral cuticularisation compared to *T. septentrionalis*. *Thalassoalaimus septentrionalis spissus* was later raised to the level of species by *Wieser (1953)* who noted additional morphological differences in the position of the cephalic setae and precloacal supplements compared to *T. septentrionalis*. More recently, *Leduc & Gwyther (2008)* mistakenly listed *T. spissus* and *T. septentrionalis spissus* as synonyms of *T. septentrionalis* due to misinterpretation of *Wieser (1953)*. As a result, *T. spissus* was left out of the identification key of *Martelli et al. (2017)*. Here, a key to valid species of *Thalassoalaimus* which includes *T. spissus*, one additional species described since Martelli et al.'s key was published (*T. crassicaudatus Huang, Sun & Huang, 2017*), and *Thalassoalaimus duoporus* sp. nov., is presented.

### *Thalassoalaimus duoporus* sp. nov.

Figures 9–11, Table 5
urn:lsid:zoobank.org:act:340CAF5E-BCA0-4809-AD13-FF0A7FF4903C

**Type locality:** Hikurangi Margin off east coast of New Zealand's North Island, Mungaroa seep site, 2,019–2,070 m water depth, sediment depth 0–5 cm, *R. Tangaroa* voyage TAN1904, stations 23 (41.9403°S, 175.3157°W) and 17 (41.9354°S, 175.3076°W).

**Type material**: Holotype male (NIWA 154935), and one paratype female (NIWA 154936), collected in July 2019.

**Measurements:** See Table 5 for detailed measurements.

**Description:** Male. Body colourless, tapering slightly towards anterior extremity. Cuticle smooth. A few, short, sparse somatic setae present, approximately 2 μm long. Metanemes and epidermal glands not observed. Labial region slightly rounded, not offset from body contour, with thickened cuticle. Six inner and six outer labial setae in close proximity and of same length, ca. 0.5 cbd; cephalic setae located slightly posterior to amphids (1.3 cbd from anterior body extremity), slightly longer than inner and outer labial setae, approximately 0.4 cbd long. Ocelli absent. Amphidial fovea horseshoe-shaped with slightly cuticularized outline; amphidial aperture markedly smaller, oval-shaped, situated near anterior edge of amphidial fovea. Buccal cavity narrow, uniformly tubular, not cuticularized and without teeth. Pharynx muscular, cylindrical, gradually widening posteriorly and with swollen posterior bulb; pharyngeal lumen not cuticularised. Pharyngeal glands visible in posterior pharyngeal bulb but their orifices indistinct. Cardia well-developed, 13 μm long, partially surrounded by intestine. Nerve ring surrounding pharynx slightly anterior to middle of pharynx length. Secretory-excretory system present, pore surrounded by thickened cuticle and with short seta slightly anterior to opening, located 74 μm (ca. 3.2 cbd) from anterior body extremity; ventral gland located immediately anterior to pharyngeal bulb.

Reproductive system diorchic with opposed and outstretched testes; anterior testis located to the left of intestine with relatively large, globular sperm, ca. 7–10 × 8–12 μm, posterior testis located to the right of intestine, with smaller spherical sperm, ca. 2.5 ×

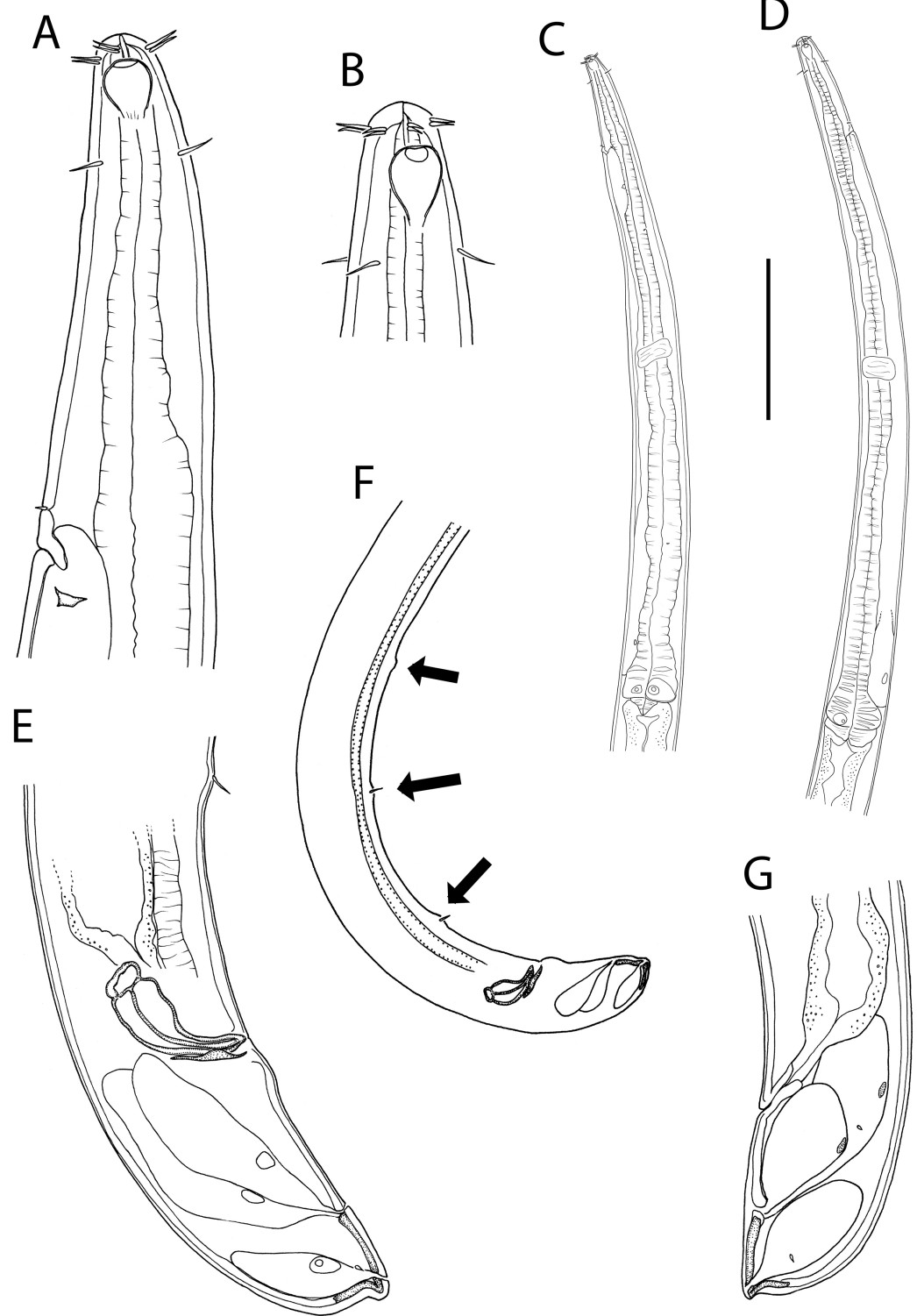

**Figure 9** *Thalassoalaimus duoporus* **sp. nov.** (A) Female anterior body region; (B) female cephalic region; (C) male pharyngeal body region; (D) female pharyngeal body region; (E) male posterior body region; (F) male posterior body region; (G) female posterior body region. Scale bar: A = 25 μm, B = 22 μm, C and D = 122 μm, E = 33 μm, F = 75 μm, G = 45 μm.

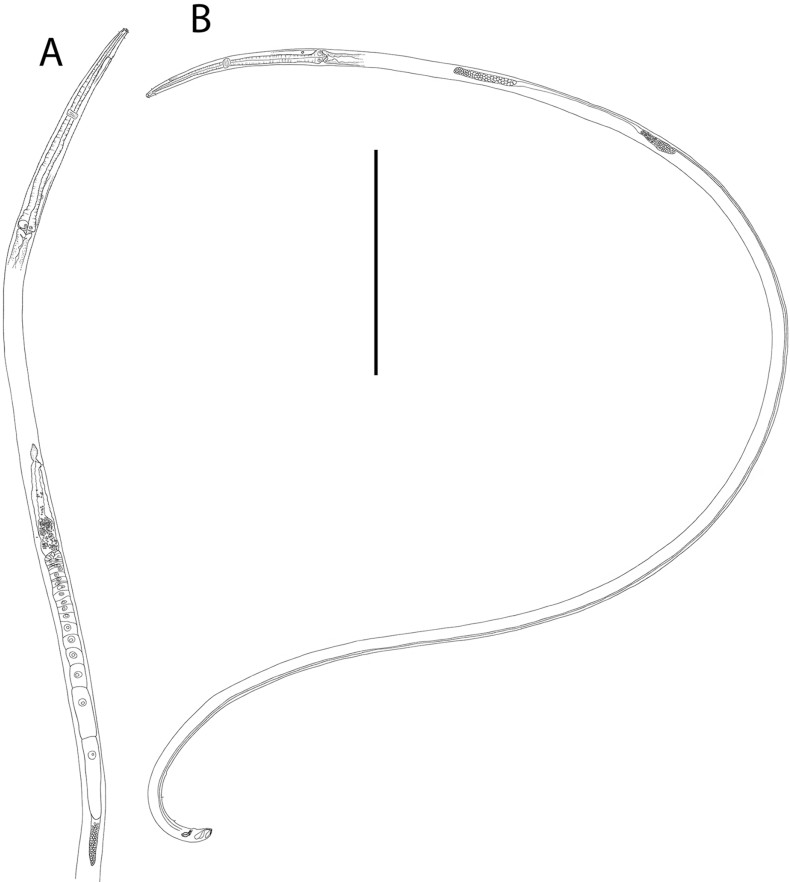

**Figure 10 _Thalassoalaimus duoporus_ sp. nov.** (A) Female anterior body region and reproductive system; (B) entire male. Scale bar: A = 500 µm, B = 610 µm.

2.5 µm. Spicules paired, symmetrical, arcuate, 0.9 cloacal body diameter long; rounded capitulum, shaft swollen proximally and blade gradually tapering distally. Gubernaculum curved, ca. half as long as spicules, with small lateral crurae partially surrounding spicules laterally. Accessory apparatus composed of three supplements consisting of raised cuticle, with two posterior-most supplements with single, 3–4 µm long seta each. Posterior-most supplement located 55 µm from cloaca; posterior-most and middle supplements 75 µm apart, middle and anterior-most supplements 66 µm apart. Tail short, conical, with caudal capsule consisting of thickened cuticle near pointed tail tip; three caudal glands connected to two separate ventral openings on distal third of tail.

Female. Similar to male. Reproductive system with single posterior reflexed ovary located to the left of intestine. Small spermatheca (or possibly gland?) present anterior to vulva. Vulva situated at approximately one fifth of body length from anterior body extremity. Vagina perpendicular, short, without cuticularisation; vaginal glands not observed.

**Diagnosis:** _Thalassoalaimus duoporus_ sp. nov. is characterised by body length 4,567–5,546 µm; anterior sensilla in 6 + 6 + 4 arrangement, inner and outer labial setae ca. 0.5 cbd long,

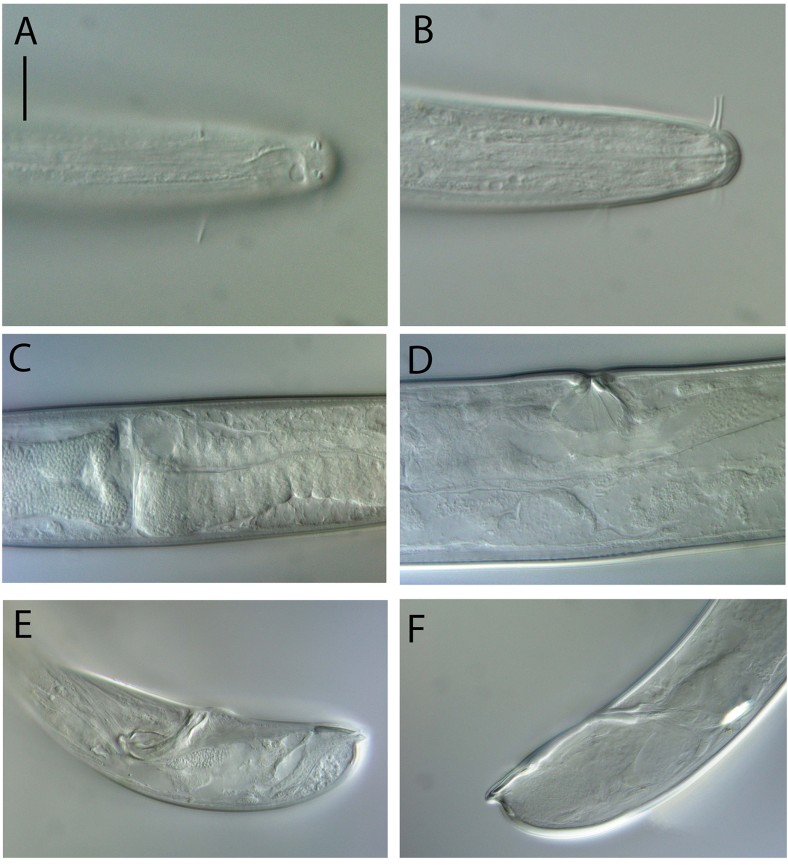

**Figure 11 *Thalassoalaimus duoporus* sp. nov.** Light micrographs. (A and B) Male anterior body region showing anterior sensilla, buccal cavity and amphid; (C) junction of intestine and posterior portion of pharynx in female; (D) secretory-excretory pore and ampulla; (E) male posterior body region; (F) female posterior body region. Scale bar: A and B = 10 μm, C, E and F = 22 μm, D = 17 μm.

cephalic setae 1.3 cbd from anterior body extremity and ca. 0.4 cbd long; amphids 4–6 μm from anterior body extremity; secretory-excretory pore ca. 3.2 cbd from anterior body extremity; arcuate spicules, gubernaculum ca. half as long as spicules, three precloacal supplements almost equidistant from each other (75 and 66 μm apart); vulva located at 19% of body length from anterior body extremity; short conical tail with pointed tail tip and caudal capsule, and caudal glands connected to two separate ventral openings on distal third of tail.

**Differential diagnosis:** The new species differs from all other species of the genus except *T. pacificus Murphy, 1965* in having body length greater than 4.5 mm and more than two precloacal supplements. *Thalassoalaimus duoporus* sp. nov. differs from *T. pacificus* in having longer inner and outer labial setae (0.5 *vs*. <0.25 cbd in *T. pacificus*), longer cephalic setae (0.4 *vs*. <0.2 cbd in *T. pacificus*), vulva located further posteriorly (19 *vs*. 12.5% of body length from anterior body extremity in *T. pacificus*), and fewer precloacal supplements (three *vs*. seven supplements in *T. pacificus*).

**Table 5 Morphometrics (μm) of _Thalassoalaimus duoporus_ sp. nov.**

| | Male Holotype | Female Paratype |
|---|---|---|
| Specimen | M1 | F1 |
| L | 4,567 | 5,546 |
| a | 97 | 96 |
| b | 9 | 11 |
| c | 104 | 105 |
| c′ | 1.3 | 1.3 |
| Head diam. at cephalic setae | 16 | 16 |
| Head diam. at amphids | 12 | 12 |
| Length of outer labial setae | 5 | 5 |
| Length of cephalic setae | 6 | 5 |
| Amphidial fovea height | 8 | 7 |
| Amphidial fovea width | 7 | 7 |
| Amphidial fovea width/cbd (%) | 58 | 58 |
| Amphidial fovea from anterior end | 6 | 4 |
| Amphidial aperture height | 3 | 3 |
| Amphidial aperture width | 2 | 2 |
| Nerve ring from anterior end | 229 | 221 |
| Nerve ring cbd | 34 | 40 |
| Pharynx length | 486 | 496 |
| Pharyngeal diam. at base | 38 | 43 |
| Pharynx cbd at base | 47 | 50 |
| Max. body diam. | 47 | 58 |
| Spicule length | 33 | – |
| Gubernaculum length | 18 | – |
| Cloacal/anal body diam. | 35 | 40 |
| Tail length | 44 | 53 |
| V | – | 1,030 |
| %V | – | 19 |
| Vulval body diam. | – | 54 |

Note:
a, body length/maximum body diameter; b, body length/pharynx length; c, body length/tail length; c′, tail length/anal or cloacal body diameter; cbd, corresponding body diameter; L, total body length; V, vulva distance from anterior end of body; %V, V/total body length × 100.

**Etymology:** The species name is derived from the latin _duo_ (= two) and _porus_ (= hole) and refers to the presence of two outlets for the caudal glands in this species.

**Key to valid species of the genus _Thalassoalaimus_ based on male characters (updated from _Martelli et al. (2017)_)**

1. Tail without pointed tip.................................. _T. brasiliensis Gerlach, 1956_

- Tail with pointed tip ......................................................................... 2

2. More than two precloacal supplements, body length greater than 4.5 mm ........... 3

- Two precloacal supplements, body length less than 4.2 mm.......................... 4

3. Seven setose precloacal supplements ....................... *T. pacificus Murphy, 1965*

- Three precloacal supplements................................... *T. duoporus* sp. nov.

4. Cephalic setae longer than ½ of corresponding diameter ............................ 5

- Cephalic setae less or equal to ½ of corresponding diameter ......................... 6

5. Gubernaculum more than ½ spicule length............. *T. nestori Martelli et al., 2017*

- Gubernaculum less than ½ spicule length .............. *T. macrosmaticus Wieser, 1953*

6. Anterior sensilla configuration: 6 + 6 + 4.......................................... 7

- Anterior sensilla configuration: 6 + 4 + 4......................................... 10

7. Cephalic setae more than three times the cephalic diameter from anterior end
   ...................................................... *T. tardus de Man, 1893*

- Cephalic setae less than three times the cephalic diameter from anterior end......... 8

8. Distance between two precloacal supplements same as distance between cloaca and
   posterior-most supplement...................... *T. spissus (Allgén, 1932) Wieser, 1953*

- Distance between two precloacal supplements greater than distance between cloaca and
   posterior-most supplement........................................................ 9

9. Secretory-excretory pore approximately three corresponding body diameters from
   anterior body extremity ............................... *T. mediterraneus Vitiello, 1970*

- Secretory-excretory pore less than two corresponding body diameters from anterior body
   extremity ................................*T. crassicaudatus Huang, Sun & Huang, 2017*

10. Tail less than two cloacal body diameters long ................*T. lissus Gagarin, 2009*

- Tail more than two cloacal body diameters long........ *T. septentrionalis Filipjev, 1927*

Family Phanodermatidae *Filipjev, 1927*

**Family diagnosis: (modified from** *Smol, Muthumbi & Sharma (2014)*) Dorsolateral and ventrolateral orthometanemes, a few loxometanemes (type I), caudal filament absent. Buccal cavity small, with or without teeth, when present one small dorsal and two large ventrosublateral teeth that all point forward (teeth may be cuticularized and protrude freely in the buccal cavity or may only be tooth-like thickenings of the pharynx lumen wall). Cephalic capsule strong or weak. Outline of posterior part of pharynx wall crenate. The pharyngeal glands open immediately posterior to the buccal cavity. Secretory-excretory gland usually present and confined to the pharyngeal region. Females didelphic-amphidelphic with antidromously reflexed ovaries. Males diorchic with opposite testes, precloacal tubule present or absent. Both gonads located to left or subventral-left of intestine.

**Remarks.** *Platonova (1984)* distinguished the genera of this family based on five types of the cephalic and pharyngeal capsules (also illustrated in *Zograf, Trebukhova & Pavlyuk, 2015*). Cephalic capsules consist of thickened cephalic cuticle offset from rest of the body by a constriction and/or grooves, sometimes with 'holes' where the amphids are located and in some cases comprising two parts (anterior and posterior). The pharyngeal capsule is

the area where the pharynx is fused with the basal internal somatic cuticle, sometimes with three protrusions ('outgrowths') of pharynx tissue anteriorly.

Subfamily Crenopharynginae *Filipjev, 1934*

**Subfamily diagnosis: (modified from *Smol, Muthumbi & Sharma (2014)*)** Small buccal cavity without cuticularized teeth, instead 'muscular teeth' formed by thickening of pharyngeal wall may be present. Cephalic capsule reduced, appears as narrow ring.

Genus *Crenopharynx Filipjev, 1934*

**Generic diagnosis: (from *Shimada & Kakui (2019)*)** Pharyngeal capsule with outgrowths filled with muscular tissue at the anterior end, cephalic capsule weakly developed (type III pharyngo-cephalic complex of *Platonova (1984)*). Ocelli absent. Tail conicocylindrical in shape, filiform in cylindrical part. Spicules longer than 3.0 anal body diameters. Precloacal supplement absent.

**Type species:** *C. gracilis* (*von Linstow, 1900*) *Filipjev, 1934*
= *Anoplostoma gracile von Linstow, 1900*
= *Stenolaimus gracilis* (*von Linstow, 1900*) *Filipjev, 1927*

**Remarks.** The genus was revised *Shimada & Kakui (2019)* who provided a key to valid *Crenopharynx* species based on male characters.

*Crenopharynx crassipapilla* **sp. nov.**
Figures 12–14, Table 6
urn:lsid:zoobank.org:act:DE34A85E-B250-4FB3-AF28-91D27F753789

**Type locality:** Hikurangi Margin off east coast of New Zealand's North Island, Uruti South seep site, 1,232–1,237 m water depth, sediment depth 0–5 cm, *R. Tangaroa* voyage TAN1904, stations 67 (41.4264°S, 176.3500°W), 68 (41.4260°S, 176.3506°W) and 70 (41.4253°S, 176.3509°W).

**Type material**: Holotype male (NIWA 154937), one paratype male and two paratype females (NIWA 154938-154940), collected in July 2019.

**Measurements:** See Table 6 for detailed measurements.

**Description:** Males. Body colourless, tapering towards anterior extremity. Cuticle smooth, 4–6 μm thick in mid-pharyngeal region, 9–11 μm thick in mid-body region, and 7–10 μm thick in mid-tail region. Eight longitudinal rows of short, sparsely distributed somatic setae present, 3–4 μm long, mostly in pharyngeal and cloacal regions. Dorsolateral and ventrolateral orthometanemes (*i.e.*, metanemes parallel with longitudinal axis) and type I loxometanemes (*i.e.*, oblique relative to longitudinal axis confined to lateral cords) present; both caudal and frontal filaments appear present, but often indistinct. Type III pharyngo-cephalic complex: pharyngeal capsule with three relatively small outgrowths filled with muscular tissue at anterior end; cephalic capsule weakly developed, length ca. 16 μm, with wavy (undulating) posterior contour. Labial region slightly rounded, bearing six stout,

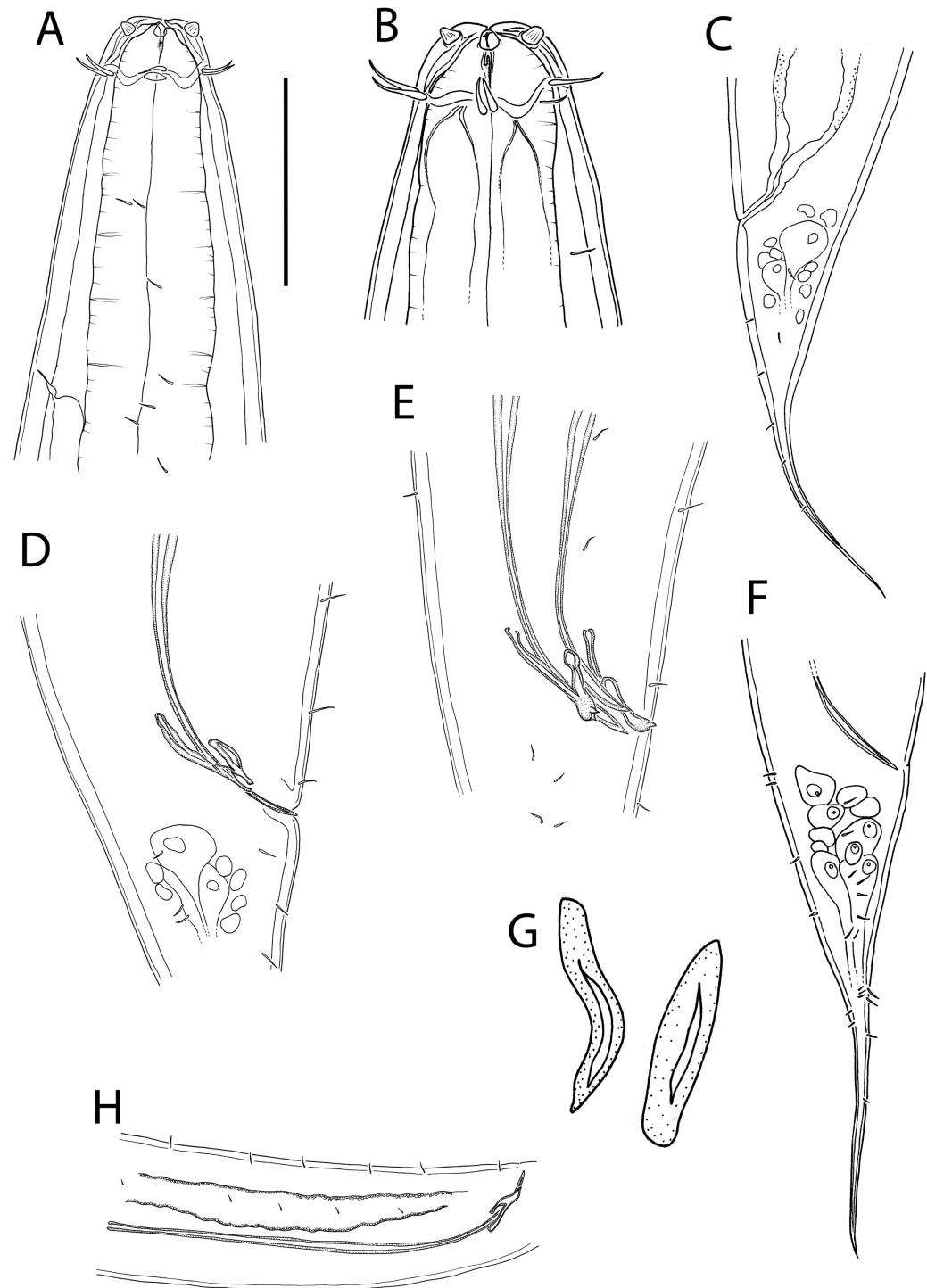

**Figure 12** ***Crenopharynx crassipapilla* sp. nov.** (A) Anterior body region of female; (B) male cephalic region; (C) female posterior body region; (D and E) male copulatory apparatus showing distal portion of spicules and gubernaculum; (F) male posterior body region; (G) sperm; (H) male copulatory apparatus showing entire spicule. Scale bar: A = 50 μm, B = 35 μm, C = 200 μm, D and E = 100 μm, F = 150 μm, G = 15 μm, H = 240 μm.

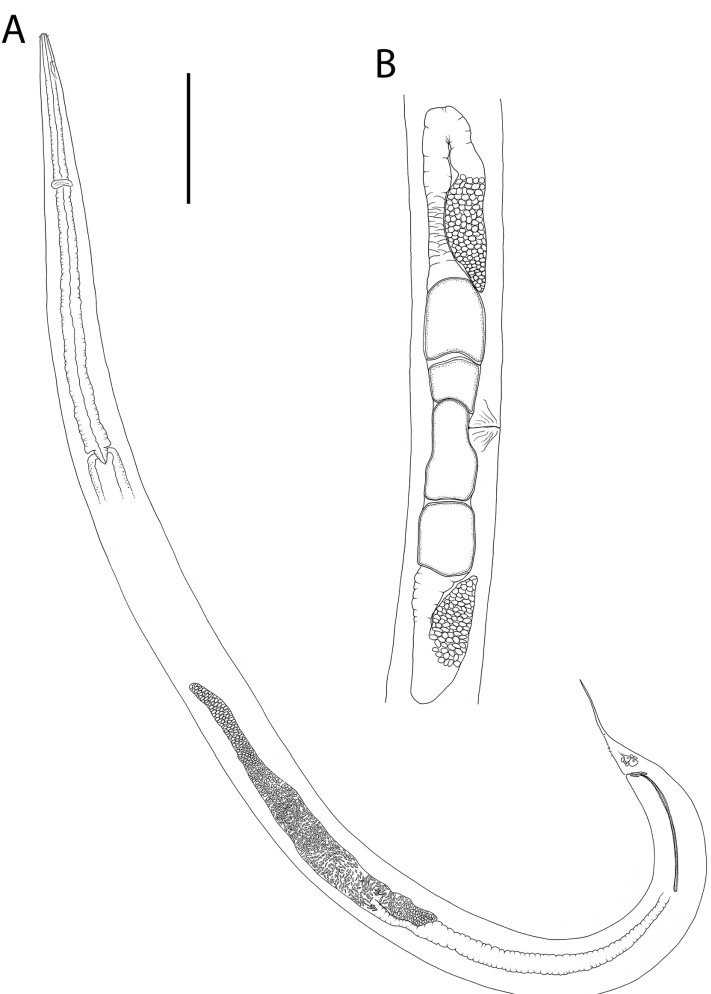

**Figure 13 *Crenopharynx crassipapilla* sp. nov.** (A) Entire male; (B) female reproductive system. Scale bar: A = 500 μm, B = 330 μm.

conspicuous inner labial sensilla 3–4 μm long and 3 μm wide at base, with 'fleshy' appearance. Circle of six outer labial setae, ca. 0.4 cbd, immediately anterior to cephalic setae of same length or slightly longer than outer labial setae. Ocelli absent. Amphidial fovea indistinct; amphidial aperture narrow, elliptical, situated immediately posterior to lateral outer labial setae. Buccal cavity narrow, funnel-shaped, with 'muscular tooth'-like fold along the ventrosublateral sectors of the pharyngeal wall. Pharynx muscular, cylindrical, gradually widening posteriorly but without posterior bulb; pharyngeal lumen not cuticularised. Pharyngeal glands not observed but pharyngeal ducts visible and with ducts apparently extending to base of buccal cavity. Cardia well-developed, ca. 30 μm long, surrounded by intestine. Nerve ring surrounding pharynx slightly posterior to one third of pharynx length. Secretory-excretory pore located ca. 1.5 cbd from anterior body extremity; ventral gland not observed.

Reproductive system diorchic with opposed and outstretched testes both located to the left of intestine. Anterior testis well-developed, with elongated or spindle-shaped sperm ca.

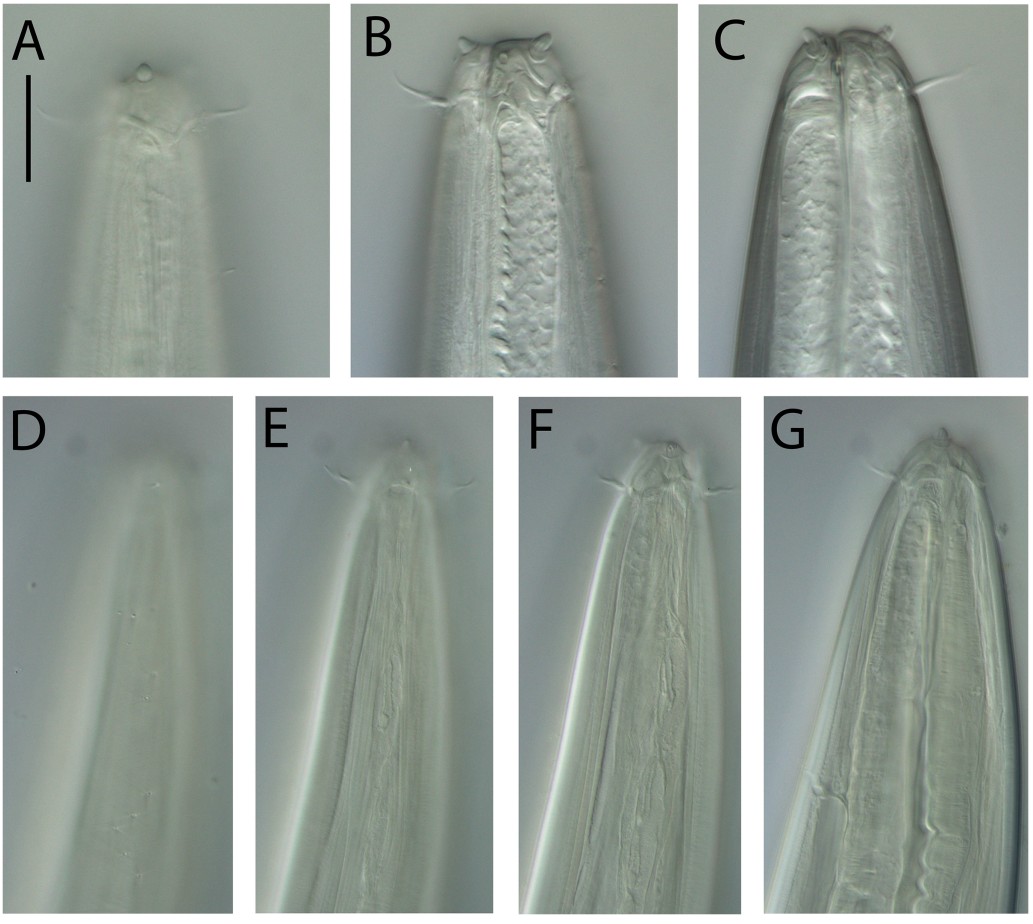

**Figure 14** *Crenopharynx crassipapilla* **sp. nov.** Light micrographs of male. (A–C) Cephalic region showing outline of cephalic capsule, anterior sensilla and buccal cavity; (D–E) Anterior body region showing body and anterior sensilla, amphid aperture and secretory-excretory pore. Scale bar: A–C = 20 µm, D–G = 28 µm.

4-5 × 15–18 µm; posterior testis much less developed, with smaller globular to spherical sperm, ca. 3 × 3–4 µm. Spicules paired, symmetrical, 5.1–5.5 cloacal body diameter long, straight proximally and curved distally, gradually tapering distally and without ventral projection or swelling. Gubernaculum paired, curved, flanking the proximal portion of spicules laterally; lateral branches extending dorso-anteriorly, and with protrusion or spines proximally (depending on viewing angle). Precloacal supplements absent. Tail of medium length, conical, with pointed tip and subventral, subdorsal and sublateral setae 7–12 µm long. Three caudal glands present, spinneret indistinct. No terminal setae.

Females. Similar to male but with fewer and shorter setae (4–8 µm) on tail. Reproductive system with two opposed reflexed ovaries both located to the left of intestine. Spermatheca not observed. Vulva situated slightly posterior to mid-body. Vagina perpendicular, without cuticularisation; vaginal glands not observed.

**Diagnosis:** *Crenopharynx crassipapilla* sp. nov. is characterised by body length 5,258–6,634 µm; conspicuous, stout inner labial sensilla, 3–4 µm long and 3 µm wide at

**Table 6 Morphometrics (μm) of *Crenopharynx crassipapilla* sp. nov.**

| | Males | | Females | |
|---|---|---|---|---|
| | Holotype | Paratype | Paratypes | |
| Specimen | M1 | M2 | F1 | F2 |
| L | 5,677 | 5,496 | 5,258 | 6,634 |
| a | 24 | 29 | 23 | 25 |
| b | 4 | 4 | 4 | 4 |
| c | 15 | 16 | 14 | 17 |
| c′ | 3.8 | 3.3 | 3.7 | 3.5 |
| Head diam. at cephalic setae | 27 | 26 | 28 | 29 |
| Head diam. at amphids | 28 | 27 | 28 | 29 |
| Length of outer labial setae | 9–10 | 10 | 10 | 11–13 |
| Length of cephalic setae | 12 | 11 | 10 | 11 |
| Amphidial aperture height | 2 | 2 | 2 | 2 |
| Amphidial aperture width | 5 | 6 | 6 | 5 |
| Amphidial aperture width/cbd (%) | 18 | 22 | 21 | 17 |
| Amphidial aperture from anterior end | 13 | 13 | 13 | 12 |
| Nerve ring from anterior end | 549 | 528 | 546 | 553 |
| Nerve ring cbd | 139 | 135 | 147 | 144 |
| Pharynx length | 1,560 | 1,460 | 1,484 | 1,524 |
| Pharyngeal diam. at base | 99 | 96 | 106 | 124 |
| Pharynx cbd at base | 225 | 176 | 216 | 245 |
| Max. body diam. | 234 | 191 | 226 | 262 |
| Spicule length | 544 | 530 | – | – |
| Gubernaculum length | 85 | 62 | – | – |
| Cloacal/anal body diam. | 99 | 104 | 103 | 111 |
| Tail length | 379 | 343 | 378 | 385 |
| V | – | – | 3,014 | 3,659 |
| %V | – | – | 57 | 55 |
| Vulval body diam. | – | – | 219 | 262 |

Note:
a, body length/maximum body diameter; b, body length/pharynx length; c, body length/tail length; c′, tail length/anal or cloacal body diameter; cbd, corresponding body diameter; L, total body length; V, vulva distance from anterior end of body; %V, V/total body length × 100.

base, inner labial setae and cephalic setae 0.4 cbd long; secretory-excretory pore ca. 1.5 cbd from anterior body extremity; spicules 5.1–5.5 cloacal body diameters, without protrusions or spines; gubernaculum paired with lateral branches extending dorso-anteriorly; tail conical, l3.5–3.8 cloacal/anal body diameters long.

**Differential diagnosis:** The new species differs from all other species of the genus in having conspicuous, stout inner labial sensilla with a 'fleshy' appearance instead of papilliform inner labial sensilla in all other species. The description of a *C. marioni* (*Southern, 1914*) *Filipjev, 1934* juvenile specimen by *Schuurmans Stekhoven & Adam (1931)* shows inner labial sensilla similar to those of *C. crassipapilla* sp. nov., however the original description of the species by *Southern (1914)* does not show or mention enlarged

inner labial sensilla. The spicular apparatus of *C. marioni* and *C. crassipapilla* differs in both the shape of the spicules (gradually tapering *vs.* suddenly tapering distally in *C. marioni*) and gubernaculum (with lateral branches *vs.* without lateral branches in *C. marioni*). Other morphological differences between the species include shorter cephalic setae (0.4 *vs.* ca. 1.0 cbd in *C. marioni*) and tail shape (conical with pointed tip *vs.* conicocylindrical with rounded tip in *C. marioni*). The stout inner labial sensilla of *C. crassipapilla* sp. nov. should not be confused with the inner labial sensilla of *C. eina Inglis, 1964*, which is characterised by inner labial papillae located on swollen bases formed by the lips, but with the papillae themselves being minute.

**Etymology:** The species name is derived from the latin *crassus* (= thick, fat, stout) and *papilla* (= nipple, bud) and refers to the unusual shape of the inner labial sensilla in this species.

**Key to valid species of the genus *Crenopharynx* based on male characters (updated from *Shimada & Kakui (2019)*)**

1. Labial sensilla stout, with 'fleshy' appearance, ca. 3–4 μm long and 3 μm wide at base ...................................................... *C. crassipapilla* sp. nov.

\- Labial sensilla papilliform ...................................................... 2

2. Spicules with projection or swelling ............................................. 3

\- Spicules without projection or swelling .......................................... 6

3. Gubernaculum with a distal branch ............................................. 4

\- Gubernaculum without branch .................................................. 5

4. Gubernaculum with a thin branch on lateral side, conical part of tail equal to cylindrical part ...................................................... *C. eina Inglis, 1964*

\- Gubernaculum with a thick branch on ventral side, conical part of tail shorter than cylindrical part ............ *C. paralepturus* (*Schuurmans Stekhoven, 1950*) *Wieser, 1953*

5. Spicules with a ventral swelling at 1/4 from distal end ............. *C. afra Inglis, 1964*

\- Spicules with a ventral swelling at distal end . *C. crassa* (*Ditlevsen, 1930*) *Filipjev, 1934*

6. Gubernaculum obviously wider than spicule ...................................... 7

\- Gubernaculum narrow, equal to spicules in width .................................. 9

7. Spicules expanded in distal part ............. *C. gracilis* (Linstow, 1900) *Filipjev, 1934*

\- Spicules suddenly or gradually tapering in distal part ............................... 8

8. Gubernaculum with distal projections on both ventral and dorsal sides .... *C. fringilla* Warwick, 1973

\- Gubernaculum simply conical ... *C. longisetae* (*Chitwood, 1936*) *Zograf, Trebukhova & Pavlyuk, 2015*

9. Spicules suddenly tapering in distal part ... *C. marioni* (*Southern, 1914*) *Filipjev, 1934*

\- Spicules gradually tapering in distal part. *C. caudata* (*Filipjev, 1927*) *Shimada & Kakui, 2019*

Family Rhabdodemaniidae *Filipjev, 1934*

**Family diagnosis: (emended from** *Holovachov & Shoshin (2014)***)** Cuticle smooth over entire body. Lateral alae absent. Body pores and epidermal glands absent. Somatic sensilla absent. Metanemes (orthometanemes) present. Labial region truncate, with six lips fused in three lobes. Inner labial sensilla small and papilliform, located on the anterior surface of the lips. Outer labial sensilla setiform; cephalic sensilla setiform, of same length or shorter than outer labial setae, may be located posterior to, at same level as, or anterior to outer labial setae. Subcephalic and cervical sensilla, deirid and ocelli absent. Amphidial aperture pore-like. Secretory-excretory system present. Excretory ampulla absent. Excretory duct opens to the exterior, somewhat anterior to the nerve ring level. Buccal cavity large and funnel-shaped: cheilostome can usually be seen as the cylindrical or funnel-shaped anterior-most part, not cuticularised; pharyngostome conoid, with strongly cuticularised lining. Anterior part of pharyngostome with three pairs of odontia located in the midventral and dorsosublateral sectors, posterior part of pharyngostome with three large onchia, one dorsal and two ventrosublateral. Three sets of three denticles may be present at base of stoma. Pharyngeal tubes absent. Pharynx uniformly cylindrical; muscularized with uniformly distributed myofilaments. Dorsal and two subventral gland orifices penetrate pharyngeal lumen somewhat posterior to the stoma base. Cardia ovoid, glandular. Pseudocoelomocytes are often present at the level of the pharyngo-intestinal junction. Female reproductive system didelphic-amphidelphic, with equally developed branches; ovaries reflexed antidromously. Spermatheca axial. Vulva equatorial, transverse. Vagina straight; pars proximalis vaginae encircled by single sphincter muscle. Male reproductive system monorchic, anterior testis outstretched. Spicules symmetrical, arcuate; gubernaculum present. Copulatory apparatus composed of a row of midventral precloacal papilliform sensilla, few in number. Two caudal glands present, their nuclei are incaudal. Spinneret functional.

**Remarks:** Rhabdodemaniidae is a monotypic family with a single genus. The placement of this family within the Triplonchida by *De Ley & Blaxter (2002,* *2004)* was not based on any morphological or molecular phylogenetic evidence, and is considered provisional by *Holovachov & Shoshin (2014)* due to the lack of information on the male copulatory musculature. So far, the only morphological feature thought to be unique for the Triplonchida is the modification of the spicule protractor muscles into two capsule-like structures that surround the anterior part of each spicule and which appear to squeeze out the spicules, as opposed to a musculature which retracts the spicules as in the Enoplida (*De Ley & Blaxter, 2002*).

The 18S ribosomal RNA phylogeny of *Smythe (2015)* suggests that the Rhabdodemaniidae should be placed within the order Enoplida rather than within the Triplonchida. Examination of available sequence data shows that the rare nucleotide transition A → G and transversion G → Y in the evolutionarily conserved loops of Hairpins 35 and 48 of the SSU rRNA gene both occur in *Rhabdodemania*, which also support its placement within the Enoplida (Z. Zhao, 2023, personal communication, *Zhao, Li & Buckley (2012)*).

Genus *Rhabdodemania Baylis & Daubney, 1926*
= *Demania Southern, 1914*
= *Pendulumia Allgén, 1954*
= *Conistomella Schuurmans Stekhoven, 1942*

**Genus diagnosis:** same as family diagnosis.
Type species: *R. major* (*Southern, 1914*) *Baylis & Daubney, 1926*
= *Demania major Southern, 1914*

**Remarks.** The genus was reviewed by *Platonova (1974)*. *Boucher (1971)* synonymised *R. scandinavica Schuurmans Stekhoven, 1946* with *R. minor* (*Southern, 1914*) *Baylis & Daubney, 1926* and considered *R. striata* Schiulz, 1932 as *species inquirenda*. Here, I consider *R. brevicaudata* (*Schuurmans Stekhoven, 1942*) *Lorenzen, 1981* and *R. obtusicauda* (*Allgén, 1954*) *Hope & Murphy, 1972* as *species inquirendae* because they were described based on female specimens only. *Hope (1988)* provides detailed electron microscopy observations of the feeding apparatus of *R. minima Chitwood, 1936*, and provides terminology (which is used here) for the designation of buccal cavity structures.

Species of this genus have been placed into two main groups based on the presence of either one or two circles of setiform sense organs by *Boucher (1971)*. Here, I suggest to modify this scheme so that species are either ascribed to group 1, characterised by the cephalic setae located at same level as the outer labial setae (6 + 10 arrangement), group 2, characterised by the cephalic setae located anteriorly to the outer labial setae (6 + 4 + 6 arrangement), or group 3, characterised by the cephalic setae located posteriorly to the outer labial setae (6 + 6 + 4 arrangement).

**Group 1. *Rhabdodemania* species with cephalic sense organs in 6 + 10 arrangement**
*R. angustissima Platonova, 1974*
*R. calycolaimus Schuurmans Stekhoven & Mawson, 1955*
*R. deconincki Platonova & Tchesunov, 1989*
*R. dura Inglis, 1966*
*R. eduntula Platonova, 1974*
*R. gracilis* (*Ditlevsen, 1918*) *Baylis & Daubney, 1926*
= *Demania gracilis Ditlevsen, 1918*
*R. hexonchia Platonova, 1974*
*R. latifaux Platonova, 1974*
*R. marisalbi Platonova, 1974*
*R. mediterranea Boucher, 1971*
*R. minima Chitwood, 1936*
*R. nancyae Inglis, 1964*
*R. orientalis Platonova, 1974*
*R. pontica Platonova, 1965*

**Group 2. *Rhabdodemania* species with cephalic sense organs in 6 + 4 + 6 arrangement**
*R. brigittae Jensen, 1976*
*R. coronata Gerlach, 1952*
*R. imer Warwick & Platt, 1973*
*R. illgi Wieser, 1959*
*R. laticauda* (*Ditlevsen, 1926*) *Baylis & Daubney, 1926*
= *Demania laticauda Ditlevsen, 1926*
*R. major* (*Southern, 1914*) *Baylis & Daubney, 1926*
= *Demania major Southern, 1914*
*R. minor* (*Southern, 1914*) *Baylis & Daubney, 1926*
= *Demania minor Southern, 1914*
= *R. scandinavica Schuurmans Stekhoven, 1946*

**Group 3. *Rhabdodemania* species with cephalic sense organs in 6 + 6 + 4 arrangement**
*R. koltchaki Platonova & Kulangieva, 1995*
*R. kudinovae Platonova & Tchesunov, 1989*

***Rhabdodemania zealandiaensis* sp. nov.**
Figures 15–17, Table 7
urn:lsid:zoobank.org:act:92315D4C-F101-4D3E-85FE-44C35309B4D1

**Type locality:** Hikurangi Margin off east coast of New Zealand's North Island, Uruti South seep site, 1,227 m water depth, sediment depth 0–5 cm, *R. Tangaroa* voyage TAN1904, station 61 (41.4251°S, 176.3510°W).

**Type material**: Holotype male (NIWA 154942), three paratype males and three paratype females (NIWA 154941), collected in July 2019.

**Measurements:** See Table 7 for detailed measurements.

**Description:** Males. Body colourless, cylindrical, tapering slightly towards anterior extremity. Cuticle smooth, 4–5 µm thick in mid-pharyngeal region, 5 µm thick in mid-body region, and 6–7 µm thick in mid-tail region. Short somatic setae present, 1–2 µm long, sparsely distributed throughout body. Metanemes (orthometanemes) present; frontal filaments ca. 2–3 cbd long, caudal filament short. Labial region slightly rounded, consisting of three lips with longitudinal buccal striae, each bearing two inner labial papillae. Circle of six outer labial setae, ca. 0.3 cbd long, located slightly anterior to four slightly shorter cephalic setae, ca. 0.2 cbd long. Ocelli absent. Amphidial aperture pore-like, often indistinct, situated slightly posterior to level of cephalic setae setae and slightly displaced ventrally relative to the lateral outer labial seta. Amphidial nerve usually visible, with sinuous contour extending from amphidial aperture posteriorly towards nerve ring. Buccal cavity large, funnel-shaped; cheilostome wide, only slightly cuticularised, pharyngostome conoid, with strongly cuticularised lining. Anterior part of pharyngostome with three pairs of odontia located in the midventral and dorsosublateral sectors; posterior part of pharyngostome with three onchia, one large dorsal onchium located anteriorly to

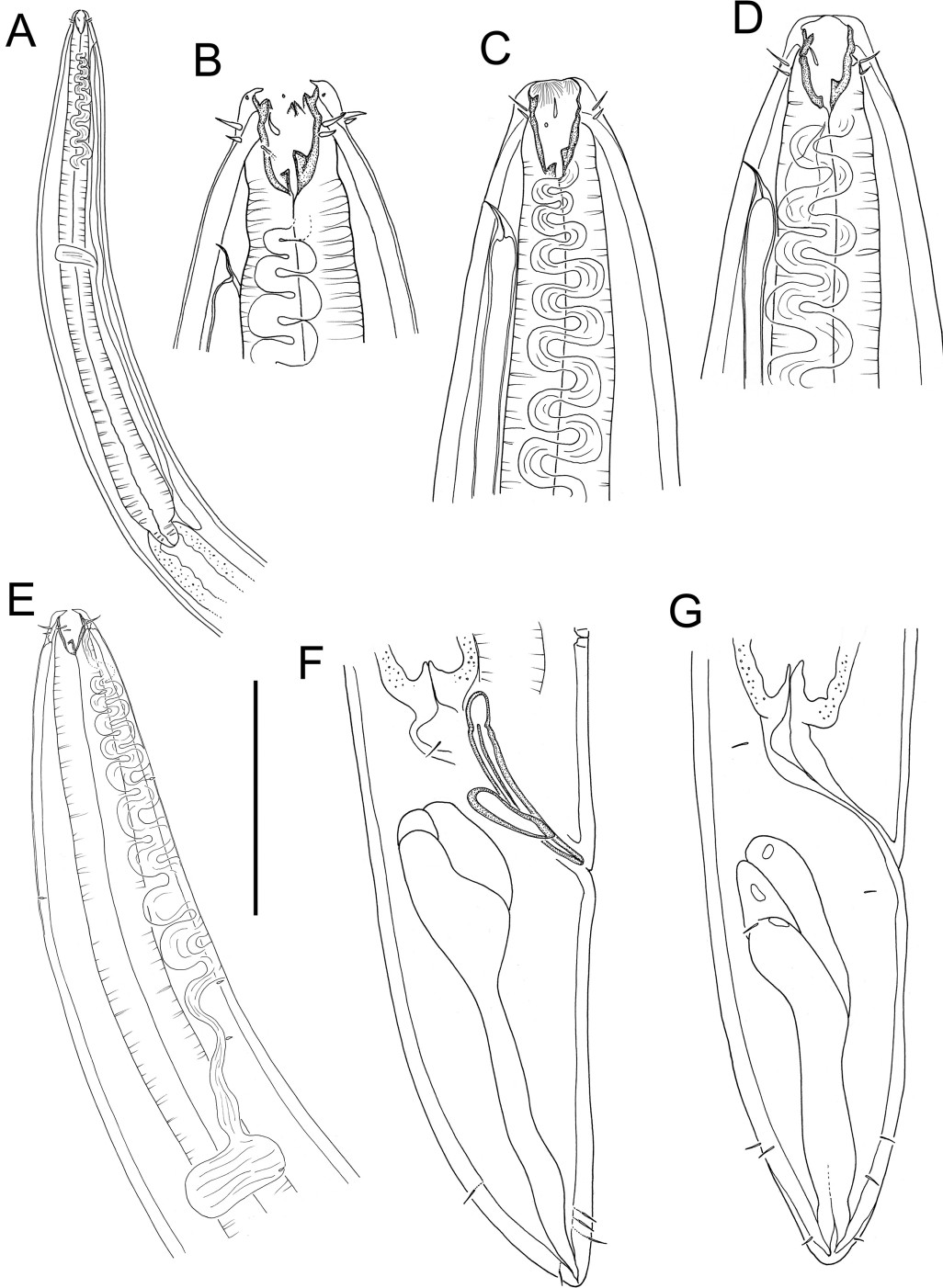

**Figure 15 *Rhabdodemania zealandiaensis sp. nov.*** (A) Male pharyngeal region; (B) female cephalic region; (C and D) male cephalic region; male anterior pharyngeal region; (F) male posterior body region; (G) female posterior body region. Scale bar: A = 250 µm, B and D = 36 µm, C = 48 µm, E = 85 µm, F and G = 80 µm.

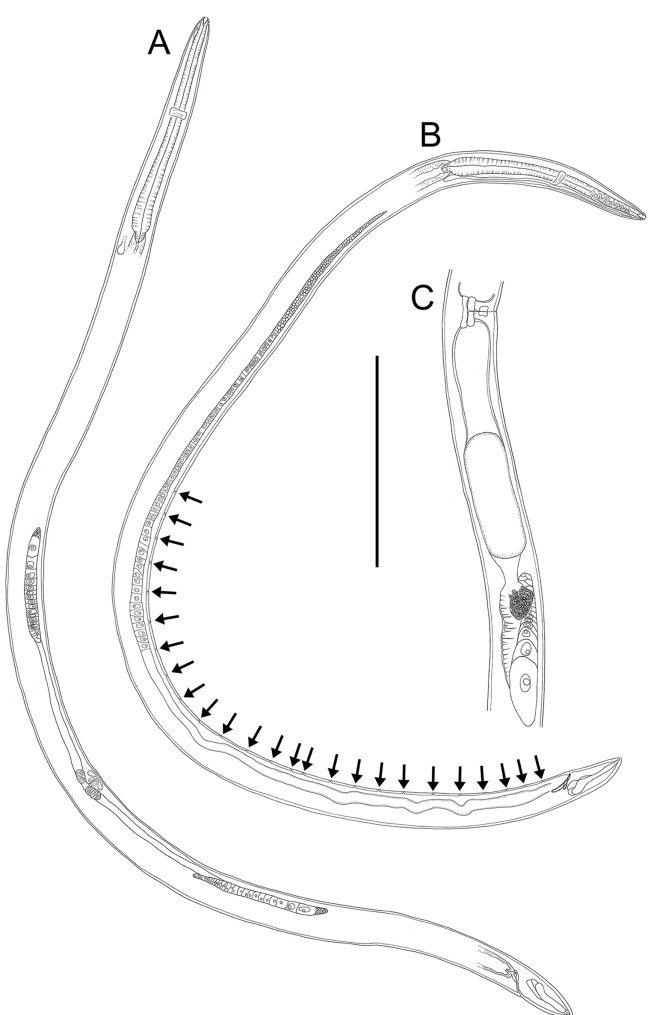

**Figure 16 *Rhabdodemania zealandiaensis sp. nov.*** (A) Entire female; (B) entire male; (C) posterior portion of female reproductive system. Scale bar: A = 500 μm, B = 550 μm, C = 380 μm.

two smaller ventrosublateral onchia at base of stoma. Denticles not observed. Pharynx muscular, cylindrical, gradually widening posteriorly but without posterior bulb; pharyngeal lumen not cuticularised. Pharyngeal glands and ducts not observed. Cardia well-developed, ca. 20–30 μm long, surrounded by intestine. Nerve ring surrounding pharynx slightly anterior to middle of pharynx length. Secretory-excretory system present; pore located ca. 0.7–0.8 cbd from anterior body extremity; ventral gland located at level of cardia.

Reproductive system diorchic with single anterior outstretched testis located ventrally relative to intestine. Sperm spherical ca. 4 × 4 μm (as measured in female uterus). Spicules paired, symmetrical, 0.8–1.0 cloacal body diameter long, almost straight and slightly curved, with weakly developed capitulum and tapering distally. Gubernaculum slightly longer than half of spicule length, flanking the proximal portion of spicules laterally and forming rounded apophyses proximally. Twenty-two to twenty-five pore-like precloacal

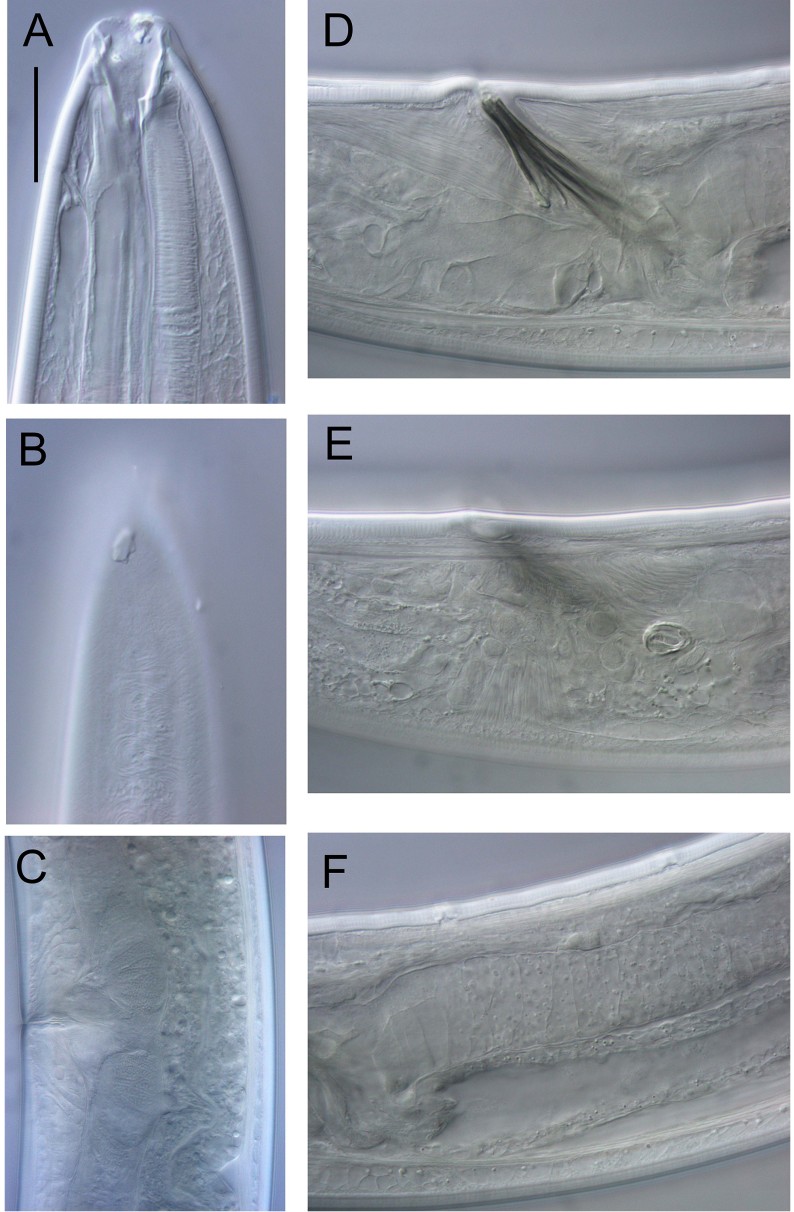

**Figure 17** *Rhabdodemania zealandiaensis sp. nov.* Light micrographs. (A) Male anterior body region showing buccal cavity, pharynx and secretory-excretory pore; (B) male anterior body region showing cuticle surface and amphidial nerve; (C) vulva; (D and E) male copulatory apparatus; (F) posterior portion of male intestine, vas deferens and copulatory supplements. Scale bar: A and B = 20 μm, C = 40 μm, D–F = 32 μm.

supplements present, beginning ca. 1.0 cloacal body diameters anteriorly to cloaca and mostly equidistant (38–45 μm) from each other; each pore surrounded by thickened cuticle and connected to small gland. Tail short, conical, with sparse subventral and subdorsal setae (4–6 μm long) near posterior extremity; three caudal glands and spinneret present.

**Table 7 Morphometrics (μm) of *Rhabdodemania zealandiaensis* sp. nov.**

| | Males | | | | Females | | | |
|---|---|---|---|---|---|---|---|---|
| | Holotype | Paratypes | | | Paratypes | | | |
| Specimen | M1 | M2 | M3 | M4 | F1 | F2 | F3 | F4 |
| L | 3,451 | 3,376 | 3,047 | 3,393 | 3,100 | 3,198 | 3,482 | 2,985 |
| a | 33 | 32 | 35 | 30 | 32 | 35 | 31 | 29 |
| b | 7 | 6 | 6 | 6 | 6 | 6 | 7 | 6 |
| c | 25 | 25 | 24 | 25 | 25 | 26 | 26 | 25 |
| c′ | 1.9 | 1.7 | 1.8 | 1.8 | 1.9 | 1.8 | 1.6 | 1.7 |
| Head diam. at cephalic setae | 19 | 19 | 19 | 18 | 18 | 18 | 18 | 18 |
| Length of outer labial setae | 5, 6 | 5, 6 | 5, 6 | 5, 5 | 5, 5 | 5 | 6 | 5 |
| Length of cephalic setae | 3, 4 | 4 | 3 | 3, 4 | 3 | 3 | 2 | 3 |
| Nerve ring from anterior end | 232 | 241 | 218 | 259 | 270 | 231 | 242 | 247 |
| Nerve ring cbd | 71 | 77 | 68 | 72 | 67 | 68 | 77 | 71 |
| Pharynx length | 514 | 535 | 492 | 579 | 513 | 528 | 530 | 533 |
| Pharyngeal diam. at base | 46 | 60 | 51 | 54 | 54 | 43 | 55 | 56 |
| Pharynx cbd at base | 92 | 100 | 81 | 98 | 82 | 83 | 103 | 93 |
| Max. body diam. | 105 | 105 | 88 | 112 | 96 | 91 | 114 | 102 |
| Spicule length | 70 | 70 | 64 | 61 | – | – | – | – |
| Gubernaculum length | 41 | 36 | 38 | 40 | – | – | – | – |
| Cloacal/anal body diam. | 72 | 79 | 70 | 78 | 67 | 67 | 85 | 73 |
| Tail length | 136 | 137 | 127 | 138 | 126 | 123 | 133 | 121 |
| V | – | – | – | – | 1,765 | 1,864 | 1,979 | 1,562 |
| %V | – | – | – | – | 57 | 58 | 57 | 52 |
| Vulval body diam. | – | – | – | – | 96 | 90 | 114 | 102 |

Note:
a, body length/maximum body diameter; b, body length/pharynx length; c, body length/tail length; c′, tail length/anal or cloacal body diameter; cbd, corresponding body diameter; L, total body length; V, vulva distance from anterior end of body; %V, V/total body length × 100.

Females. Similar to males. Reproductive system with two opposed reflexed ovaries both located ventrally relative to intestine. Vulva situated slightly posterior to mid-body. Vagina perpendicular, without cuticularisation; vaginal glands present. Base of each uterus sometimes surrounded by sphincter muscle; spermathecae not observed.

**Diagnosis:** *Rhabdodemania zealandiaensis* sp. nov. is characterised by body length 2,985–3,482 μm; anterior sensilla in 6 + 6 + 4 arrangement, cephalic setae slightly shorter (0.2 cbd) than outer labial setae (0.3 cbd); buccal cavity with three pairs of equal odontia, one large dorsal onchium located anteriorly to two smaller ventrosublateral onchia at base of stoma; spicules almost straight, 0.8–1.0 cloacal body diameter long, 22–25 pore-like precloacal supplements mostly equidistant (38–45 μm) from each other; vulva located 52–58% body length from anterior body extremity; short conical tail.

**Differential diagnosis:** The new species is very similar to *R. kudinovae*, which was described by *Platonova & Tchesunov (1989)* from the South Atlantic, west of the South

Orkney Islands (December 1971; water depth = 807–912 m; 57.117°S, 26.683°W). The main difference between the two species is the absence of precloacal supplements in *R. kudinovae*, whereas 22–25 relatively conspicuous supplements are present in *R. zealandiaensis* sp. nov. Other inconsistencies between the two species include: the slightly shorter outer labial setae in *R. zealandiaensis* sp. nov. (5–6 *vs.* 7–8 μm in *R. kudinovae*) and the narrower buccal cavity (8–9 *vs.* 11–13 μm in *R. kudinovae*). Although the values overlap, the position of the vulva in *R. kudinovae* is much more variable (%V = 46–73) than in *R. zealandiaensis* (%V = 52–58).

**Etymology:** The species name refers to the submerged continent of Zealandia, where the type specimens were discovered.

## DISCUSSION

The present study builds on previous work providing descriptions of six new nematode species of the class Chromadorea from the same Hikurangi Margin seep localities *(Leduc, 2023)* Of the twelve macrofaunal nematode species described so far from the Hikurangi Margin seep sites, all are new to science, which shows that a high proportion of the nematode diversity on New Zealand's continental margin remains to be described. *Metacylicolaimus catherinae* sp. nov. represents the first record of the genus for the New Zealand Exclusive Economic Zone and for the deep sea globally (>200 m depth; *Miljutin et al., 2010*). *Halalaimus talaurinus* sp. nov., *Thalassoalaimus duoporus* sp. nov. and *Crenopharynx crassipapilla* sp. nov. are only the second species of their respective genera to be described/recorded from the region (*Leduc & Gwyther, 2008*). *Oncholaimus adustus* sp. nov. is the eight species of the genus to be recorded from New Zealand waters (*Ditlevsen, 1930*; *Leduc & Gwyther, 2008*; *Leduc, 2009*).

*Rhabdodemania zealandiaensis* sp. nov. was among the most abundant and widespread species found at the Hikurangi Margin seep sites (D. Leduc, 2022, unpublished data) and a few specimens were also found in a study of meiofaunal nematode communities on Chatham Rise, a submarine ridge south of Hikurangi Margin (*Leduc et al., 2012*). This species was much less common on Chatham Rise, which is probably at least partly due to differences in methodologies; the core samples used in this study were relatively large (ca. 70 cm$^2$) and were processed using a coarse mesh (300 mm), unlike in the Chatham Rise study which used relatively small core samples (<10 cm$^2$) and a finer (45 mm) mesh (*Leduc et al., 2012*). It is possible that this species has a preference for seep environments due to the greater food availability, however it does not seem to be exclusively found in seeps.

Based on the limited evidence available, the discovery of several new nematode species at the Hikurangi Margin seep sites suggests that there is little affinity between nematode seep communities in New Zealand and elsewhere. These findings are in line with the high variability in nematode community observed to date among different seep locations (*Vanreusel et al., 2010b*). Ongoing work on the taxonomy, ecology and distribution of nematode communities at the Hikurangi Margin seep sites will help determine spatial patterns in abundance and species distributions in more detail, including the identification of any species with affinities for methane seepage.

## ACKNOWLEDGEMENTS

I thank the science party participants of voyage TAN1904 and the officers and crew of *R. Tangaroa*. I am very grateful to Dr. Alexei Tchesunov and Dr. Vadim Mokievsky for their help with Russian translation and obtaining original species descriptions. I also thank two anonymous reviewers for providing constructive criticisms on the manuscript.

## ABBREVIATIONS

| | |
|---|---|
| **a** | Body length/maximum body diameter |
| **b** | Body length/pharynx length |
| **c** | Body length/tail length |
| **c′** | Tail length/anal or cloacal body diameter |
| **L** | Total body length |
| **n** | Number of specimens |
| **V** | Vulva distance from anterior end of body |
| **%V** | V/total body length $\times$ 100 |

### Funding

Funding was provided by NIWA's Oceans Centre Research Programme 'Protecting Marine Biodiversity' and the HYDEE research programme through the funding agencies: Ministry for Business, Innovation and Employment (MBIE) (project C05X1708) and NIWA (under sub-contract GNS-MBIE00079). The funders had no role in study design, data collection and analysis, decision to publish, or preparation of the manuscript.

### Grant Disclosures

The following grant information was disclosed by the authors:
NIWA: GNS-MBIE00079.
Ministry for Business, Innovation and Employment: C05X1708.

### Competing Interests

The author declares that they have no competing interests.

### Author Contributions

- Daniel Leduc conceived and designed the experiments, performed the experiments, analyzed the data, prepared figures and/or tables, authored or reviewed drafts of the article, and approved the final draft.

### Field Study Permissions

The following information was supplied relating to field study approvals (*i.e.*, approving body and any reference numbers):

The New Zealand's Ministry for Primary Industries approved this study (Permit 666).

## Data Availability

The raw data are available in Tables 1–7.

All specimens are held in the NIWA Invertebrate Collection (Wellington, New Zealand).

*Metacylicolaimus catherinae* sp. nov.: Holotype male (NIWA 154925), two paratype males and four paratype females (NIWA 154926-154927).

*Oncholaimus adustus* sp. nov.: Holotype male (NIWA 154928), two paratype males and four paratype females (NIWA 154929-154931).

*Halalaimus talaurinus* sp. nov.: Holotype male (NIWA 154932), one paratype male and three paratype females (NIWA 154933-154934).

*Thalassoalaimus duoporus* sp. nov.: Holotype male (NIWA 154935), and one paratype female (NIWA 154936).

*Crenopharynx crassipapilla* sp. nov.: Holotype male (NIWA 154937), one paratype male and two paratype females (NIWA 154938-154940).

*Rhabdodemania zealandiaensis* sp. nov.: Holotype male (NIWA 154942), three paratype males and four paratype females (NIWA 154941).

## New Species Registration

The following information was supplied regarding the registration of a newly described species:

Publication LSID: urn:lsid:zoobank.org:pub:288A67E3-5436-4F9A-990F-DCFF3E49EE35

*Crenopharynx crassipapilla*: urn:lsid:zoobank.org:act:DE34A85E-B250-4FB3-AF28-91D27F753789

*Halalaimus talaurinus*: urn:lsid:zoobank.org:act:DE927E2B-B780-40DB-AE04-4EE038F5AFF1

*Metacylicolaimus catherinae*: urn:lsid:zoobank.org:act:F2B1591E-C796-4333-AA0B-D526F0FBD9B3

*Oncholaimus adustus*: urn:lsid:zoobank.org:act:4019665C-3216-4225-903B-93D61916F5B1

*Rhabdodemania zealandiaensis*: urn:lsid:zoobank.org:act:92315D4C-F101-4D3E-85FE-44C35309B4D1

*Thalassoalaimus duoporus*: urn:lsid:zoobank.org:act:340CAF5E-BCA0-4809-AD13-FF0A7FF4903C

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
