# Peer review of "Six new species of free-living nematodes (Nematoda: Enoplida) from deep-sea cold seeps on Hikurangi Margin, New Zealand"

_PeerJ, doi:10.7717/peerj.14867_

## Round 0.1 · original submission · Minor Revisions

The reviewers provided important comments, and I ask that you carefully consider them when revising the manuscript, as well as respond to their suggestions in the cover letter when resubmitting.

Please clarify the taxonomic position of the species Rhabdodemania kudinovae and correct the Latin name of species Oncholaimus ramosum.

Reviewer 1 ·

Basic reporting

The manuscript by Daniel Leduc is a very high quality paper devoted to descriptions of nematodofauna of cold seeps. The language is very clear and well understandable, figures are of a great quality, literature is sufficientl.

Experimental design

Research done within aim and scopes of the journal. Methods described with sufficient details.

Validity of the findings

All findings are new to science and are of interest to a wide range of ecologists and taxonomists.

Additional comments

I have some remarks that needs clarification.
Line 128: Subventral or ventral precolacal papillae (never tubules) often present. This is family diagnosis. And later on the line 213 supplemet structure (disk-like versus tubular). I see some contradiction between family diagnosis and species descriptions.
Line 189-190. The drawing of gubernaculum contradicts with photo and description.
Line 191. Supplement located 165-172 mkm anterior to cloaca. According to drawing supplement is closer to cloaca at the distance of ca 139 mkm.
General remarks on Oncholaimus adustus sp.nov. The buccal cavity armouring looks very unusual for genus Oncholaimus. For me this species most likely belongs to the genus Metaparoncholaimus. Moreover, the species O. problematicus and O. keinesis also most likely belongs to genus Metaparoncholaimus.
Species name Oncholaimus ramosus is listed in Nemys base as Oncholaimus ramosum. There is some confusion in species name. Initially this species was called Oncholaimium ramosum and that was correct according to Latin rules. After the species was transferred to genus Oncholaimus specific name was not changed and now the species name Oncholaimus ramosum, that is nor correct. Anyway, in the manuscript one name must be used (now there are both variants in the paper used).
Line 380. Cuticle with transverse striation begining posteriorly to amphid. At the same time at the figure 6B it begins in front of amphid.
The main problem with Rhabdodemania kudinovae is the presence of 22-25 preclocal supplements. According to Platonova & Tchesunov 1989 males have no any supplementary structures. So, the taxonomic position of this species must be clarified.

Reviewer 2 ·

Basic reporting

This is a sound and well-made paper containing descriptions of several nematode species, new for science or new for region, discovered in a peculiar milieu. Besides descriptions of new species, the paper is featured by some interesting morphological observations showing unusual or previously unknown structures in organizations of certain marine nematodes such as very thick cuticle in Halalaimus, two ventral caudal outlets in Thalassoalaimus.

Experimental design

Correct

Validity of the findings

Correct

Additional comments

The manuscript is generally well written and accurate, I have only a few minor remarks:
Line 273-274. “… blunt teeth either equal in size or with ventrosublateral tooth appearing slightly larger … “. It seems, “left” is here lost between “with” and “ventrosublateral”.
Line 293-295, fig. 4. I guess, the demanian system of Oncholaimus adustus may be described incorrectly. Normally, the uterine canal extends from the uterus posteriorly and terminates by uvette on the main canal while the main canal extends anteriorly and terminates by osmosium contacted with the intestine; the osmosium is thus situated anteriorly to the uvette. Osmosium and uvette are misrecognized? And besides, the main canal posteriorly terminates by several small ducts with outlets in perianal area. These posterior canals are not mentioned here.
Line 383 and further in the text. The cuticle of Halalaimus talaurinus looks indeed too thick. Is it not possible here to confuse somatic cuticle with entire body wall (cuticle+epidemis+muscles)? But the cuticle looks really homogeneous on photopictures.
Figure 6. I suggest to add arrows indicating cuticle pores – otherwise the tiny structures can be overlooked.
Lines 708-709. Rhabodemaniidae is a member of Enoplida, according to all morphological traits. Position of Rhabdolaimidae within Triplonchida is derived from a mistake – see Smythe, AB, 2015, Integrative and Comparative Biology, 55(2): 228-240.
References – all the nouns in German sources should be typed with a capital letter.

---

## Round 0.2 · accepted · Accept

In the revised version the author took into consideration all comments and remarks. I recommend accepting your manuscript for publication in PeerJ.

Reviewer 1 ·

Basic reporting

Authors greatly improved the manuscript and took into account all my remarks. I believe that manuscripts must be published in present state.

Experimental design

n/a

Validity of the findings

n/a

Additional comments

n/a